bioinformatics/systems biology

disease classification, gene network, modules, pathways

**Author for correspondence:**
Binghui Guo
e-mail: guobinghui@buaa.edu.cn

# Disease classification via gene network integrating modules and pathways

Zhilong Mi[1,2,4], Binghui Guo[1,2,4], Ziqiao Yin[1,2,3,4], Jiahui Li[1,2,4] and Zhiming Zheng[1,2,4]

[1]Beijing Advanced Innovation Center for Big Data and Brain Computing, Beihang University, Beijing 100191, People's Republic of China
[2]LMIB and School of Mathematics and Systems Science, Beihang University, Beijing 100191, People's Republic of China
[3]Shenyuan Honors College, Beihang University, Beijing 100191, People's Republic of China
[4]Peng Cheng Laboratory, Shenzhen, Guangdong Province 518055, People's Republic of China

ZM, 0000-0002-7811-7204; BG, 0000-0002-0540-3779

Disease classification based on gene information has been of significance as the foundation for achieving precision medicine. Previous works focus on classifying diseases according to the gene expression data of patient samples, and constructing disease network based on the overlap of disease genes, as many genes have been confirmed to be associated with diseases. In this work, the effects of diseases on human biological functions are assessed from the perspective of gene network modules and pathways, and the distances between diseases are defined to carry out the classification models. In total, 1728 diseases are divided into 12 and 14 categories by the intensity and scope of effects on pathways, respectively. Each category is a mix of several types of diseases identified based on congenital and acquired factors as well as diseased tissues and organs. The disease classification models on the basis of gene network are parallel with traditional pathology classification based on anatomic and clinical manifestations, and enable us to look at diseases in the viewpoint of commonalities in etiology and pathology. Our models provide a foundation for exploring combination therapy of diseases, which in turn may inform strategies for future gene-targeted therapy.

## 1. Introduction

Characterizing disease in the biological big data era of the twenty-first century has been of significance [1], based on not only pathological analysis and clinical syndromes but also molecular-level information, including gene data. The genes of an organism play a vital role in the regulation of cellular processes, as well as disease development. Much effort is directed at the possibility to

relate the molecular pathology of simple monogenic diseases to their associated clinical phenotypes, and extensive population studies are being carried out in finding the genes that are involved in common, multifactorial diseases [2]. Monogenic Mendelian inheritance elucidation may help to detect pathogenic mechanisms in complex diseases [3]. Involvement by multiple genes in complex diseases usually occurs but the identification of specific genes has been a challenge so far [3]. Integrative omics approaches share molecular mechanisms for Mendelian and complex diseases [4]. More and more clinical research with a large number of individuals will be required, which highlights the weight of understanding phenotype–genotype relationships in designing approaches to gene therapy [2,3].

With the advancement and deepening of disease research, more and more genes have been confirmed to be associated with one or more specific diseases. Hence, a number of institutions and organizations that collect and manage such data have emerged. The National Center for Biotechnology Information (NCBI, https://www.ncbi.nlm.nih.gov/) [5] is part of the United States National Library of Medicine, a branch of the National Institutes of Health. The NCBI holds a series of databases relevant to biotechnology and biomedicine, and is an important resource for bioinformatics tools and services. Kyoto Encyclopedia of Genes and Genomes (KEGG, http://www.genome.ad.jp/kegg/) [6] is an effort, which computerizes current knowledge on cellular processes and standardizes gene annotations, to link genomic information with higher-order functional information. Online Mendelian Inheritance in Man (OMIM, https://www.omim.org/) [7,8] is a continuously updated catalogue of human genes and genetic disorders, with a particular focus on the gene–phenotype relationship. The European Molecular Biology Laboratory (EMBL, https://www.embl.org/) groups and laboratories perform basic research in molecular biology and molecular medicine as well as training for scientists, students and visitors. Many research projects are aimed at improving the ability to diagnose, treat and prevent diseases through a better understanding of the genetic basis of diseases based on biomolecular data. For example, the Cancer Genome Atlas (TCGA) [9] is a project to catalogue genetic mutations responsible for cancer and this project was completed in 2018. The Ensembl project [10] has been aggregating, processing, integrating and redistributing genomic datasets since the initial releases of the draft human genome, with the aim of accelerating genomics research through rapid open distribution of public data. And the Cancer Gene Census (CGC) [11] is an ongoing effort to catalogue those genes which contain mutations that have been causally implicated in cancer.

Availability of genomic relationship data gives rise to a large research area of network medicine illuminating the multifactorial nature of diseases [4]. Network science offers a comprehensive toolset for exploring biological systems [12]. The regulation of gene expression is achieved through gene regulation networks which comprise genes, proteins and other interactive small molecules [13]. Many predictive regulatory/signalling models of specific phenotypes were built, such as cell cycle [14,15], apoptosis [16,17], survival [18] or inflammatory signalling [19], to help us understand and control steady states of life activities. Dávid *et al.* draw on dynamical systems theory to decipher how multiple individual functions are coordinated in health and disease, as intractable diseases such as cancer are associated with multiple individual function breakdowns, which conspire to create unhealthy phenotype combinations [20]. Wang *et al.* [21] develop a framework based on the concept of attractor networks to quantify and facilitate the study of nonlinear dynamic controllability in complex systems. This framework can help find the perturbation that drives the cancer cell state back to normal, which also proves its practicality.

Many network-based approaches are proposed in order to explore the associations between human genetic diseases and the relations between their effector genes [22–25]. Goh *et al.* [22] conduct a considerable work in which they created a network based on disease–gene associations obtained from the OMIM database. This work suggests that essential human genes are likely to encode hub proteins and are expressed widely in most tissues; however, the vast majority of disease genes are non-essential [22]. Rzhetsky *et al.* [23] construct a network where two disorders are connected based on the observed comorbidity between them. Such disease network allows us to estimate the size of putative genetic overlaps from these correlations. Particularly, the authors suggest that autism, bipolar disorder and schizophrenia share significant genetic overlaps [23]. Lee *et al.* [24] construct a bipartite human disease association network where the disease pairs are connected if mutated enzymes associated with them catalyse adjacent metabolic reactions. This network topology-based approach helps to uncover potential mechanisms that contribute to their shared pathophysiology [24].

Disease classification is the foundation for achieving precision medicine, and methods developed for disease classification are promising for revolutionizing personalized medicine and increasing our understanding of disease etiology [4]. With the complete sequence of the human genome, and with a

growing body of omic datasets in health and disease, human disease will be defined precisely with optimal sensitivity and specificity [1]. Also, it starts a rigorous analytical process that can lead to defining prognostic determinants and better-individualized therapeutic responses [1]. For a long time, cancer has been classified mainly according to the origination location in the body. However, a collaborative project called the Pan-Cancer Initiative [26], launched in 2012, plans to study cancer from another perspective. A preliminary analysis shows that cancers originating from different organs actually have something in common at the molecular level, and cancers originating from the same tissue may have very different genomic characteristics [27]. In 2018, this project has been completed. Li *et al.* take a comprehensive perspective on oncogenic processes based on Pan-Cancer Atlas analyses, giving prominence to the complex impact of genome alterations on the signalling and multi-omic profiles of human cancers as well as their influence on tumour microenvironment [9]. Hoadley *et al.* [28] show that 33 tumours analysed can be reclassified into 28 different molecular types based on their cellular and genetic composition, rather than their origin, which would inform strategies for future therapeutic development. Sanchez-Vega *et al.* [29] make an integrated analysis of genetic alterations in 10 signalling pathways across 9125 tumour samples profiled by TCGA and point out significant representation of individual and co-occurring actionable alterations in these pathways, suggesting opportunities for targeted and combination therapies.

In this work, disease classification models based on gene network are proposed, as we consider not only individual genes, but also gene network modules and pathways. The largest connected component (LCC) of the human gene network is divided into 10 modules by using the fast unfolding algorithm. We integrate disease genes, topological modules and biological pathways to assess the influence of diseases on the human body, and perform different classifications of diseases for different interests. In total, 1728 diseases collected from KEGG are divided into 12 categories by the intensity of effects on pathways, and are identified as 14 categories by the scope of effects on pathways. The number of disease categories that contain cancers is the smallest among 15 types of diseases, which suggests the similarity that cancer diseases are complex and have a great impact on pathways. Each category is a mix of several types of diseases identified based on congenital and acquired factors as well as diseased tissues and organs, which implies that the human gene network gives a perspective of disease classifications, and guides future gene-targeted therapy and combination therapy of diseases.

# 2. Methods

## 2.1. Data collection

We obtain a human gene information file (Homo_sapiens.gene_info.gz, version 4 March 2018) from NCBI (https://ftp.ncbi.nih.gov/gene/DATA/GENE_INFO/Mammalia/Homo_sapiens.gene_info.gz). In this file, 16 types of information of 60 674 human genes are listed, including GeneID, Symbol, Synonyms, type of gene, etc. Also, gene−gene interaction information (interactions.gz, version 4 March 2018) is acquired from NCBI (https://ftp.ncbi.nih.gov/gene/GeneRIF/interactions.gz). Each record is verified by one or more literatures (unique Pubmed ID) curated from publications.

KEGG PATHWAY [6] is a collection of manually drawn pathway maps representing our knowledge on the molecular interaction, reaction and relation networks. It includes not only the normal states but also the perturbed states of the biological systems, divided into seven types of pathways as follows: metabolism, genetic information processing, environmental information processing, cellular processes, organismal systems, human diseases and drug development. We select 317 human pathways apart from drug development pathways which include gene information.

Diseases are viewed as perturbed states of the biological system in KEGG DISEASE [6,30]. Each disease is represented by a list of known disease genes, any known environmental factors at the molecular level, diagnostic markers and therapeutic drugs, which may reflect the underlying molecular system. Diseases are divided into 15 primary classifications: cancers, immune system diseases, nervous system diseases, cardiovascular diseases, respiratory diseases, endocrine and metabolic diseases, digestive system diseases, urinary system diseases, reproductive system diseases, musculoskeletal diseases, skin diseases, congenital disorders of metabolism, congenital malformations, other congenital disorders and other diseases. In total, 1728 diseases (with known disease genes) of 67 secondary classifications are screened out (https://www.kegg.jp/kegg-bin/get_htext?htext=br08402_gene.keg).

## 2.2. Module partition based on the fast unfolding algorithm

The module reflects the local characteristics of individual behaviours in the network and their interrelationships. The modules in the research network play a vital role in understanding the structure and function of the entire network, and can help us analyse and predict the interaction between the elements of the entire network. A key step was taken when Girvan and Newman popularized graph-partitioning problems by introducing the concept of modularity [31]. In its original definition, an unweighted and undirected network that has been partitioned into communities has modularity

$$Q = \frac{1}{2m} \sum_{i,j} \left[ A_{ij} - \frac{k_i k_j}{2m} \right] \delta(c_i, c_j),$$

where $A$ is the adjacency matrix of the network, $m$ is the total number of edges, and $k_i = \sum_j A_{ij}$ is the degree of node $i$. The indices $i$ and $j$ run over the $N$ nodes of the graph.

The fast unfolding algorithm is a multistep method based on a local optimization of modularity in the neighbourhood of each node and implemented as follows [32]:

---

**Algorithm 1.** Fast unfolding algorithm.

---

1: Assign a different module to each node of the network.

2: **repeat**

3: **repeat**

4: **for** $i$=1 to $n$ **do**

5: **for** $j$ s.t. $A_{ij}$=1 **do**

6: Evaluate $\Delta Q_{ij}$ by removing $i$ from its module and by placing it in the module of $j$.

7: **end for**

8: Place node $i$ in the module of $k$ for which $\Delta Q_{ik}$ is maximum and positive.

9: **end for**

10: **until** no gain of $Q$ can be achieved.

11: Replace modules by supernodes.

12: Give the weights of links between new nodes by the sum of the weights of links between nodes in the corresponding two modules.

13: **until** no further improvement can be achieved.

---

## 2.3. Adjusted cosine similarity measures the correlation of gene distribution deviation between pathways

To assess the overlap of function pathways and topological modules, we get the proportion vector $\lambda^i = (\lambda_1^i, \lambda_2^i, \ldots, \lambda_{10}^i)$ for each pathway $i$ ($i = 1, 2, \ldots, 317$) in 10 modules. As each module has a different number of genes, we apply the adjusted cosine similarity to measure the correlations of pathway genes distribution in network modules

$$\text{ACS}(i, j) = \frac{\langle \lambda^i - \overline{\lambda}, \lambda^j - \overline{\lambda} \rangle}{|\lambda^i - \overline{\lambda}| \cdot |\lambda^j - \overline{\lambda}|},$$

where $\overline{\lambda} = (1/17274)(3560, 3269, 2541, 1820, 1750, 1639, 1521, 573, 414, 187)$ is the proportion vector of modules in LCC and represents the average proportion of randomly selected genes. In this way, $\text{ACS}(i, j)$ measures the correlation of gene distribution deviation between pathway $i$ and $j$.

## 2.4. The influence on pathways by diseases

In the biological literature, there is a tacit assumption that these three concepts are interrelated: cellular components that form a topological module have closely related functions, thus they also correspond to

a functional module, and a disease is a result of the breakdown of some particular functional modules [32,33]. It is considered to be right that disease genes affect pathway functions through topology modules.

The number of diseases is represented by $N_d = 1728$, the number of modules is represented by $N_m = 10$, and the number of pathways is represented by $N_p = 317$. Each disease $k$ ($k = 1, 2, \ldots, N_d$) is represented by disease genes group $D_k = \{g_1^k, g_2^k, \ldots, g_{n_k}^k\}$. Each topological module $M_l$ ($l = 1, 2, \ldots, N_m$) is represented by genes group $M_l = \{g_1^m, g_2^m, \ldots, g_{n_l}^m\}$, and each functional pathway $P_t$ ($t = 1, 2, \ldots, N_p$) is represented by genes group $P_t = \{g_1^t, g_2^t, \ldots, g_{n_t}^t\}$. To weight the influence on pathways by diseases, we first define the access efficiency from each disease within topological modules by

$$\text{AE}(D_k, M_l) = \sum_{g_i \in D_k} \frac{|M_l|}{\sum_{g_j \in M_l} d_{ij}} \delta(g_i, M_l),$$

where $|M_l| = n_l$ is the size of $M_l$, and $d_{ij}$ is the length of the shortest path between $g_i$ and $g_j$ in the LCC. $\delta(g_i, M_l) = 1$ has the value 1 for all $g_i$ in $M_l$ and the value 0 for all $g_i$ not in $M_l$. $\text{AE}(D_k, M_l)$ describes the summation of the closeness centrality of disease genes within the module $M_l$. The more disease genes within module $M_l$, the more significant $M_l$ plays a role in developing the disease.

Then we define the relevance of modules and functional pathways by jaccard similarity coefficient as follows:

$$\text{JSC}(M_l, P_t) = \frac{|M_l \bigcap P_t|}{|M_l \bigcup P_t|},$$

where $|M_l \cap P_t|$ is the size of intersection of $M_l$ and $P_t$, and $|M_l \cup P_t|$ is the size of union of $M_l$ and $P_t$. $\text{JSC}(M_l, P_t)$ describes similarity between finite gene groups $M_l$ and $P_t$. This measure is selected mainly because it provides an intuitive way to characterize the set similarity.

The impact score on function pathways by disease $D_k$ is defined as follows:

$$\text{IS}(D_k, P_t) = \sum_{l=1}^{N_m} \text{AE}(D_k, M_l) \cdot \text{JSC}(M_l, P_t)$$

and

$$\text{IS}(D_k) = (\text{IS}(D_k, P_1), \text{IS}(D_k, P_2), \ldots, \text{IS}(D_k, P_{N_p})),$$

where $\text{IS}(D_k, P_t)$ is the impact score on pathway $P_t$, and $\text{IS}(D_k)$ is the impact score vector on pathways by disease $D_k$.

To classify diseases by the intensity of effects on pathways, we use the normalized vector $\text{IS}^N(D_k)$ as the impact score vector of disease $D_k$, where

$$\text{IS}^N(D_k, P_t) = \frac{\text{IS}(D_k, P_t) - \min_s(\text{IS}(D_k, P_s))}{\max_s(\text{IS}(D_k, P_s)) - \min_s(\text{IS}(D_k, P_s))}$$

and

$$\text{IS}^N(D_k) = (\text{IS}^N(D_k, P_1), \text{IS}^N(D_k, P_2), \ldots, \text{IS}^N(D_k, P_{N_p})).$$

The distance between two diseases is defined by the Euclidean distance to describe the difference of intersity of effects on pathways, as follows:

$$\text{distance}^N(D_i, D_j) = \sqrt{\sum_{k=1}^{N_p} (\text{IS}^N(D_i, P_k) - \text{IS}^N(D_j, P_k))^2}.$$

To classify diseases by the scope of effects on pathways, we use the binary vector $\text{IS}^B(D_k)$ as the impact score vector of disease $D_k$, where

$$\text{IS}^B(D_k, P_t) = \begin{cases} 1 & \text{if } \text{IS}(D_k, P_t) \geq \dfrac{\sum_{s=1}^{N_p} \text{IS}(D_k, P_s)}{N_p} \\[3em] 0 & \text{if } \text{IS}(D_k, P_t) < \dfrac{\sum_{s=1}^{N_p} \text{IS}(D_k, P_s)}{N_p} \end{cases}$$

**Table 1.** Classification of interactions according to the species to which the gene involved belongs. There are 289 946 human–human gene interactions in which 17 309 human genes are involved. There are 31 736 interactions between 7590 human genes and 4960 non-human genes. Twenty-two withdrawn genes are involved in 33 interactions. Data are obtained from NCBI (interactions.gz, version 4 March 2018).

| type of interactions | no. interactions | no. genes involved |
| --- | --- | --- |
| human–human gene interactions | 289 946 | 17 309/17 309 |
| human–nonhuman gene interactions | 31 736 | 7590/4960 |
| human–withdrawn gene interactions | 33 | 29/22 |

and

$$\mathrm{IS}^B(D_k) = (\mathrm{IS}^B(D_k, P_1), \mathrm{IS}^B(D_k, P_2), \ldots, \mathrm{IS}^B(D_k, P_{N_p})).$$

$\mathrm{IS}^B(D_k, P_t)$ put emphasis on the pathways which are affected beyond average. The distance between two diseases is defined by the Manhattan distance to describe the proportion of different pathways affected by two diseases, as follows:

$$\mathrm{distance}^B(D_i, D_j) = \frac{\sum_{k=1}^{N_p} |\mathrm{IS}^B(D_i, P_k) - \mathrm{IS}^B(D_j, P_k)|}{N_p}.$$

## 2.5. Classification of diseases

Given a partition $P = \{\mathcal{P}_1, \mathcal{P}_2, \ldots, \mathcal{P}_s\}$ of diseases, we construct a symmetric matrix $\mathrm{Distance}_{s \times s}$, where $\mathrm{Distance}(i, i)$ is the average distance between diseases in $\mathcal{P}_i$ and $\mathrm{Distance}(i, j)$ is the average distance between diseases in $\mathcal{P}_i$ and $\mathcal{P}_j$. Let

$$\mathrm{Difference}^P = \frac{1}{s} \sum_{i=1}^{s} \mathrm{Distance}(ii) - \frac{1}{s(s-1)} \sum_{i \neq j}^{s} \mathrm{Distance}(ij)$$

represent the difference of average distances of diseases within partitions to average distances of diseases between partitions.

Using Ward.D2 method [34], we get a hierarchical clustering of diseases by distance. A given dendrogram is regarded as a series of partitions $\{P_1, P_2, \ldots, P_{N_d-1}\}$ in the order of merge. We consider partitions $\{P_r, P_{r+1}, \ldots, P_{N_d-1}\}$ in which every disease is merged at least once, and cut the dendrogram at $P_m$, $r \leq m \leq N_d - 1$ where the corresponding $\mathrm{Difference}^{P_m}$ is the minimum.

# 3. Results

Different from previous works such as classifying diseases according to the gene expression data of patient samples [28], or understanding disease gene and phenotype associations by constructing a bipartite graph consisting of diseases and genes [22], we assess the effects of diseases on human biological functions and define the distances between diseases to carry out the classification models from the perspective of gene network modules and pathways. The genes and interactions are collected from NCBI to construct the human gene network. Then the human pathways and diseases associated with human genes are collected from KEGG, which are considered together with gene network modules to develop the disease classification models. Two disease classification models are proposed for different interests.

## 3.1. Genes and interactions: underlying framework of the human gene network

We obtain human gene information from NCBI in which 16 types of information of 60 674 human genes are listed, including GeneID, symbol, synonyms, type of gene, etc. Also, gene–gene interaction information is acquired from NCBI (see details in Methods). Each record is verified by one or more literatures (unique Pubmed ID) curated from publications. See in table 1, totally 321 715 interactions including at least one human gene are screened out; specifically, 17 430 human genes are involved in at least one interaction, in which 17 309 human genes are recorded in 289 946 interactions to interact with human genes and 7590 human genes are recorded in 31 736 interactions to interact with 4960 non-human genes from 83 species.

**Table 2.** Genes and interactions of 25 small connected components in human – human gene interactions. Seventeen single-gene connected components, six double-gene connected components, two triple-gene connected components are listed. Data are obtained from NCBI (interactions.gz, version 4 March 2018).

**(a) self-interacting single-gene connected components**

| gene X | | |
| --- | --- | --- |
| Entrez ID | symbol | interaction |
| 265 | AMELX | X – X |
| 359 | AQP2 | |
| 1178 | CLC | |
| 1733 | DIO1 | |
| 6276 | S100A5 | |
| 8547 | FCN3 | |
| 22 947 | DUX4L1 | |
| 25 769 | SLC24A2 | |
| 26 103 | LRIT1 | |
| 26 257 | NKX2-8 | |
| 55 586 | MIOX | |
| 55 894 | DEFB103B | |
| 64 123 | ADGRL4 | |
| 128 674 | PROKR2 | |
| 152 404 | IGSF11 | |
| 256 130 | TMEM196 | |
| 390 648 | OR4F6 | |

**(b) two modes of interactions in double-gene connected components**

| gene X | | gene Y | | |
| --- | --- | --- | --- | --- |
| Entrez ID | symbol | Entrez ID | symbol | interaction |
| 778 | CACNA1F | 57 010 | CABP4 | X – Y |
| 5623 | PSPN | 64 096 | GFRA4 | |
| 10 800 | CYSLTR1 | 57 105 | CYSLTR2 | |
| 1259 | CNGA1 | 9187 | SLC24A1 | X – Y, Y – Y |
| 1524 | CX3CR1 | 6376 | CX3CL1 | |
| 2676 | GFRA3 | 9048 | ARTN | |

**(c) two modes of interactions in triple-gene connected components**

| gene X | | gene Y | | gene Z | | |
| --- | --- | --- | --- | --- | --- | --- |
| Entrez ID | symbol | Entrez ID | symbol | Entrez ID | symbol | interaction |
| 57 758 | SCUBE2 | 80 274 | SCUBE1 | 222 663 | SCUBE3 | X – X, X – Y, Y – Y, Y – Z, Z – Z |
| 80 834 | TAS1R2 | 80 835 | TAS1R1 | 83 756 | TAS1R3 | X – Z, Y – Z |

Thirty-three interactions are omitted because there existed genes withdrawn by NCBI. The collection of 17 309 human genes (nodes) and 289 946 interactions (undirected edges) is considered as the human gene network. The LCC of the human gene network possesses 17 274 genes and 289 913 interactions. Another 25 connected components are listed in table 2, each of which occupies no more than three genes. Degree distribution of the LCC approximates a power-law distribution with $\gamma = 1.696$ (figure 2a).

## 3.2. Pathways and diseases: normal and perturbed states of certain gene combinations

We select 317 human pathways which include gene information from KEGG (see details in Methods). For each of the six types of pathways, one subject to significant influence by diseases is selected as example to illustrate the subnetwork consisting of pathway genes (figure 1). In total, 6547 of 7409 human pathway genes belong to the LCC (figure 2c). The number of pathways in which one gene involved has a power-law distribution (figure 2b), and $\gamma = 1.966$. It suggests that minority genes involve in a large number of pathways to participate in cellular activities. In the 85 metabolism pathways, majority genes do not interact with non-human genes. However, in another 232 human pathways, most genes interact with non-human genes (figure 2d,e).

As expected, human pathway genes that interact with both human and non-human genes (component IV) are more crucial. The degree of human genes which interact with both human and non-human genes is much larger than that of human genes which only interact with human genes. Specially, component IV occupies many hubs (table 3), for the reason that the median degree as well as average degree and maximum degree is much larger than that of components I, II, III. Besides, the genes in component IV interact with more non-human genes and species, which shows genetic diversity.

For example, TRIM25 is the maximum-degree gene, and its protein is a bona fide RNA binding protein associated with many proteins involved in RNA metabolism and that interacted with numerous coding and non-coding transcripts [35]. TRIM25 plays a key role in the RIG-I signalling pathway, which is a cytosolic pattern recognition receptor that senses viral RNA [36]. Additionally, TRIM25 is involved in normal development and diseases in association with the estrogen response [37]. Moreover, interactions with 589 non-human genes of 23 non-human species including mammals and viruses provide a large understanding of UBC. UBC gene is one of the two stress-regulated polyubiquitin genes (UBB and UBC) in mammals and plays a key role in maintaining cellular ubiquitin levels under stress conditions [38,39]. Ubiquitination has been associated with protein degradation, DNA repair, cell cycle regulation, kinase modification, endocytosis and regulation of other cell signalling pathways [40–42]. Cells require either UBB or UBC for survival [43]. Furthermore, MAPK1 and MAPK3 are involved in 98 and 97 human gene pathways, respectively, which are the most. The proteins encoded by MAPK1 and MAPK3 are involved in a wide variety of cellular processes such as proliferation, differentiation, transcription regulation and development [44].

In total, 1728 diseases (with known disease genes) of 15 primary classifications are screened out in KEGG DISEASE (see details in Methods). See in table 3 that the proportion of disease genes is larger in pathway genes (components III and IV) than non-pathway genes (components I and II), and disease genes appear in pathways involved in more diseases. The average degree of disease genes is larger than that of all genes; however, most disease genes are not hubs, resulting from low median degree, which is consistent with the idea that majority of disease genes are non-essential and do not encode hub proteins [22].

## 3.3. Module partition: bridge between individual genes and pathways

Modularity proposed by Newman is one measure of the structure of networks [31]. The module reflects the local characteristics of individual behaviours in the network and their interrelationships. The modules in the research network play a vital role in understanding the structure and function of the entire network, and can help us analyse and predict the interaction between the elements of the entire network. Biological networks exhibit a high degree of modularity. Trying to understand the network-based position of disease genes, Barabási et al. have reviewed three modularity concepts: topological modules, functional modules and disease modules [32].

In this part, the LCC is considered. The fast unfolding algorithm proposed by Blonde et al. in 2008 is recognized as one of the fastest and accurate non-overlapping community discovery algorithms, especially for networks of unprecedented sizes [45]. The LCC is divided into 10 modules. Most hubs are divided into different modules to play a role in the process of implementing some function. For example, in module 1, ELAVL1 has been implicated in a variety of biological processes and has been linked to a number of diseases, including cancer. It is highly expressed in many cancers, and could be potentially useful in cancer diagnosis, prognosis and therapy. In module 2, mutations in APP have been implicated in autosomal dominant Alzheimer's disease and cerebroarterial amyloidosis. Besides, EGFR is a cell surface protein that binds to epidermal growth factor and associates with cell proliferation. In module 3, tumour suppressor gene TP53 and ubiquitin gene UBC are associated with cell cycle regulation, apoptosis, senescence, DNA repair, protein degradation or changes in metabolism.

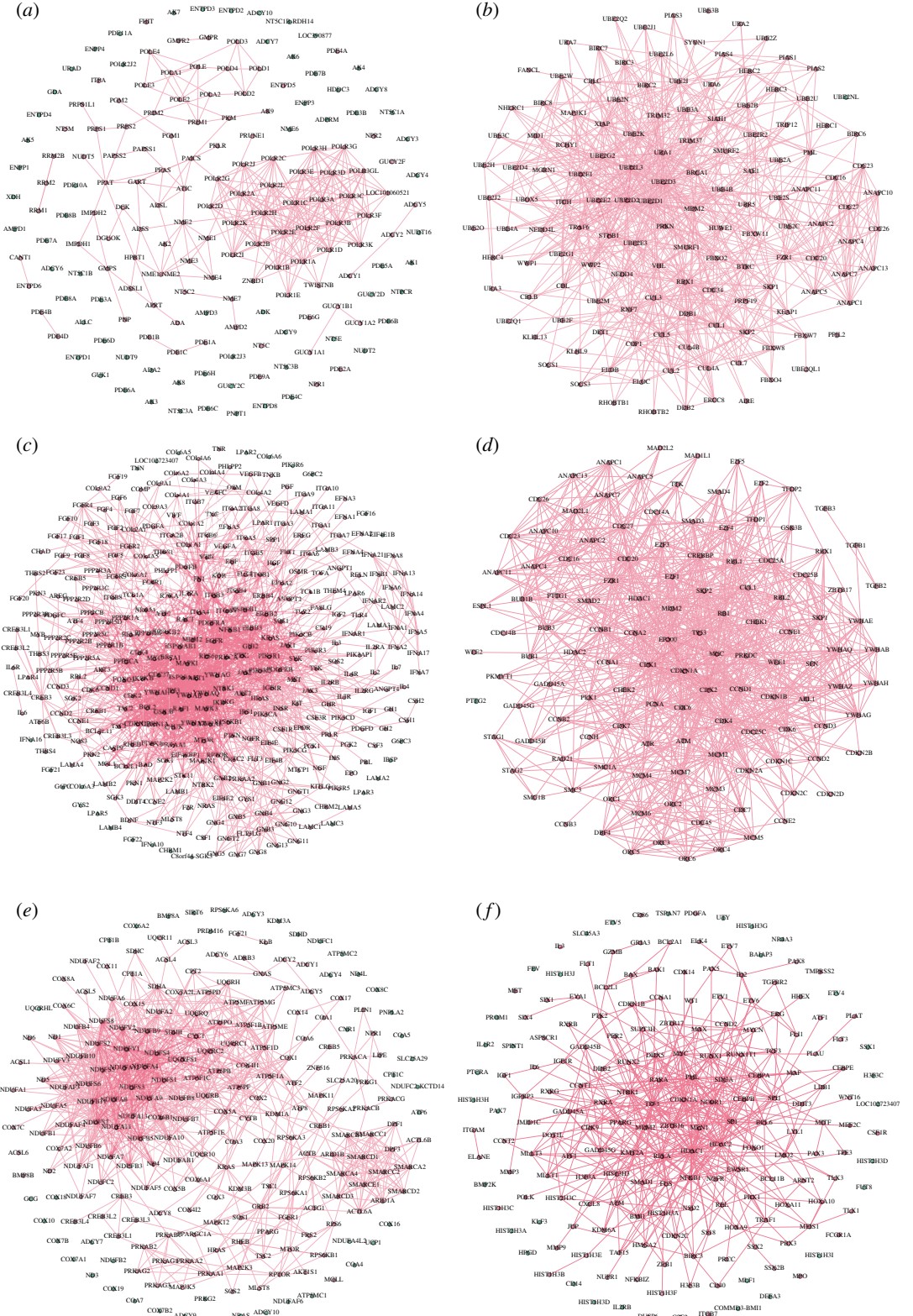

**Figure 1.** For each of the six types of pathways, we select one subject to significant influence by diseases as an example to illustrate the subnetwork consisting of pathway genes, in which isolated points are illustrated in green, while connected points are illustrated in red. (*a*) Subnetwork of purine metabolism pathway genes. (*b*) Subnetwork of ubiquitin mediated proteolysis pathway genes. (*c*) Subnetwork of PI3K-Akt signalling pathway genes. (*d*) Subnetwork of cell cycle pathway genes. (*e*) Subnetwork of thermogenesis pathway genes. (*f*) Subnetwork of microRNAs in cancer pathway genes.

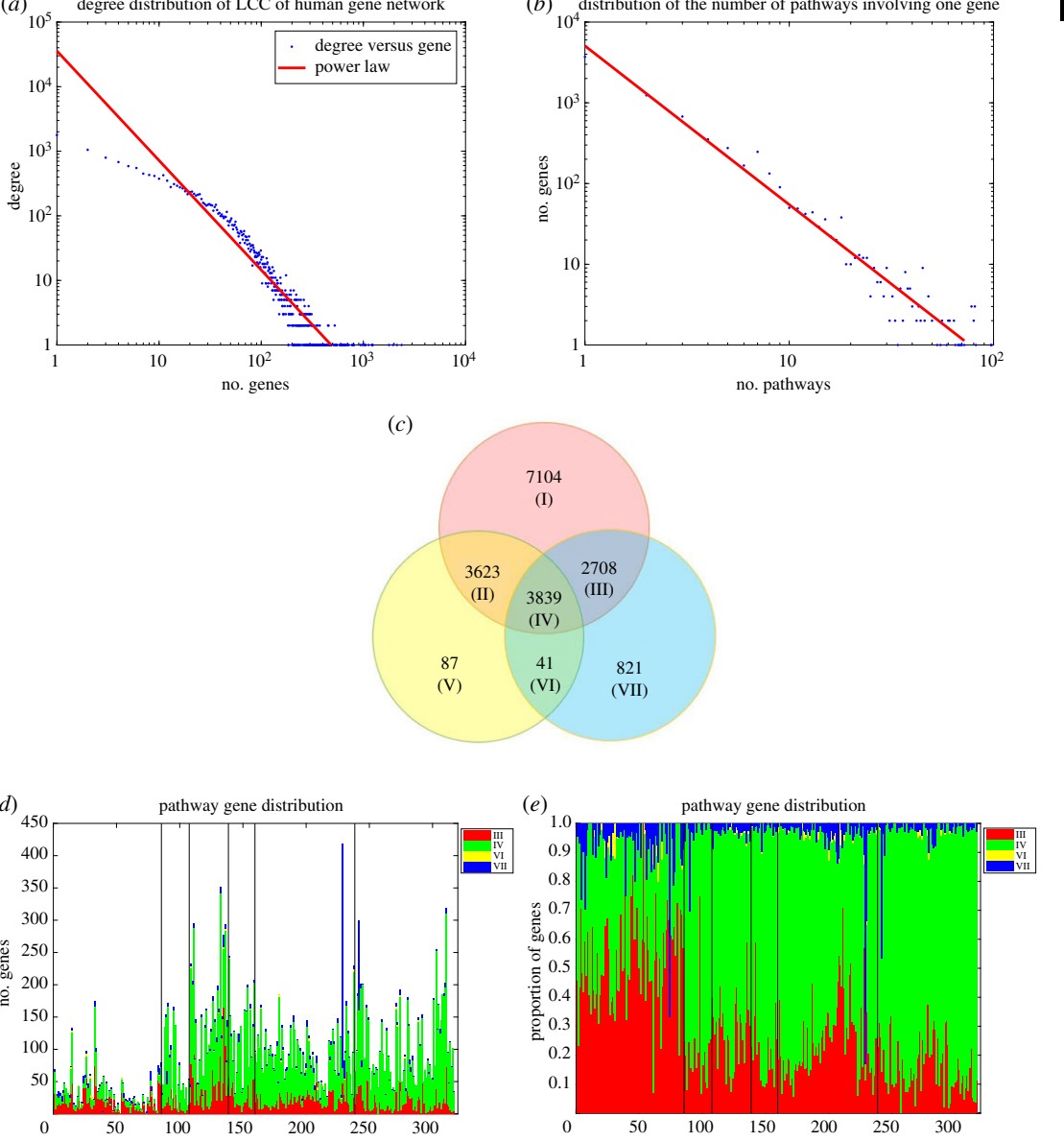

**Figure 2.** (*a*) Degree distribution of the LCC has a power-law distribution with $\gamma = 1.696$. (*b*) The number of pathways in which one gene involved has a power-law distribution with $\gamma = 1.966$. (*c*) Venn diagram of the LCC genes (17 274, red), human genes interacting with non-human genes (7590, yellow), and human pathway genes (7409, blue). The diagram is divided into seven parts. (*d*) The number of four kinds (III,IV,VI,VII) of genes in 317 human pathway genes. The number of metabolism pathway genes is fewer than that of other five types of pathways. (*e*) The proportion of four kinds (III,IV,VI,VII) of genes in 317 human pathway genes. The proportion of genes that interact with both human and non-human genes in metabolism pathways is less than that of other five types of pathways.

To assess the overlap of function pathways and topological modules, we get the proportion vector $\lambda^i = (\lambda_1^i, \lambda_2^i, \ldots, \lambda_{10}^i)$ for each pathway $i$ ($i = 1, 2, \ldots, 317$) in 10 modules, and apply the adjusted cosine similarity (see details in Methods) to measure the correlations of pathway genes distribution in network modules. Figure 3 shows the hierarchical cluster of the correlations between 317 pathways, which yields six groups. Most metabolism pathways get together in Group 1 and Group 3, most genetic information processing pathways get together in Group 4, while the other four types of pathways have more diverse groups (Groups 2, 5, 6). This means that vital differences exist between metabolism as well as genetic information processing pathways and another four types of pathways. The metabolism pathway genes mainly distribute in modules 1, 2 and 7, with above-average proportion in modules 1, 7 and 8. The genetic information processing pathway genes mainly distribute in modules 1, 3 and 4, with above-average proportion in modules 3 and 4. While another four types of pathway genes are mainly in module 2 as well as 1 and 3, with above-average proportion in modules 1, 2, 3 or 8 (table 4).

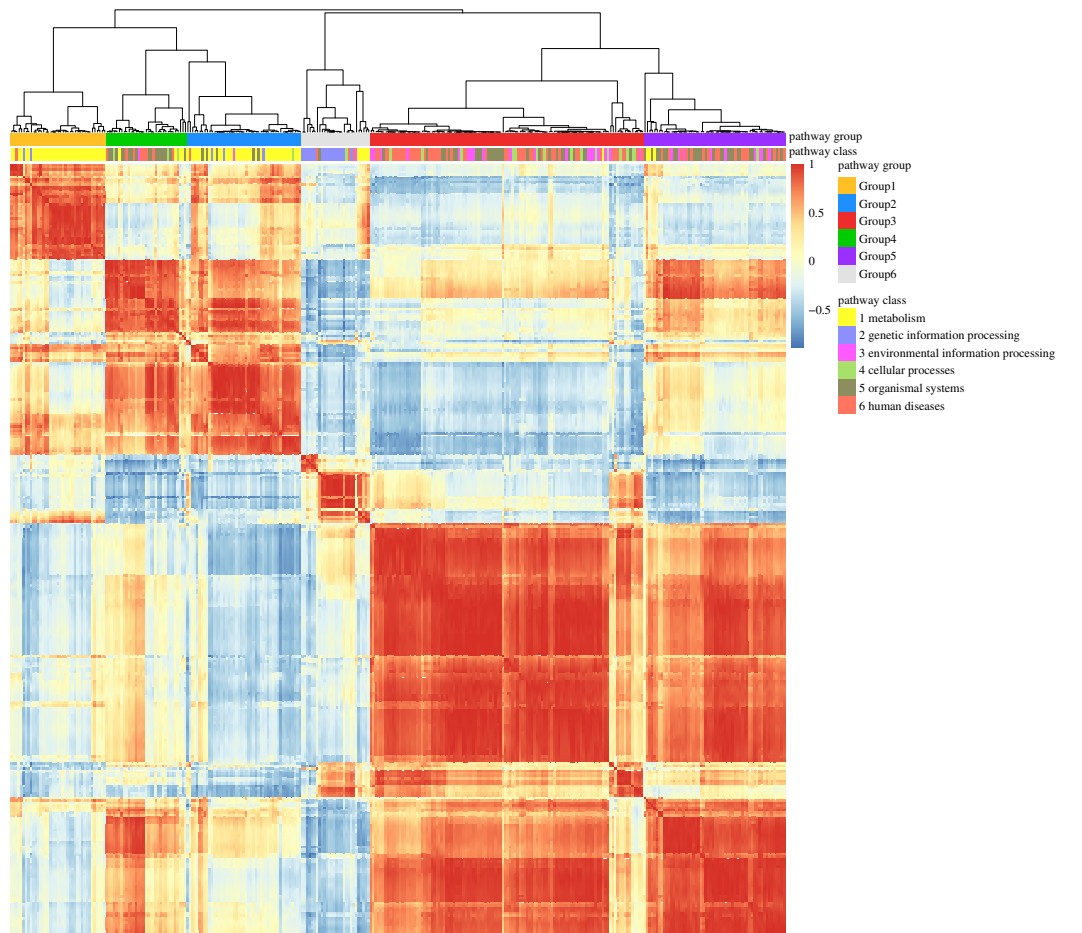

**Figure 3.** The hierarchical cluster of the correlations between 317 pathways yielded six groups. For pathway *i* and *j*, adjusted cosine similarity of proportion vectors in 10 modules is used to measure the correlation. Most metabolism pathways got together in Group 1 and Group 3, most genetic information processing pathways got together in Group 4, while four other types of pathways had more diverse groups (Groups 2,5,6).

**Table 3.** Statistical indicators of human genes and disease genes in four types of genes (I, II, III, IV). The proportion of disease genes is larger in pathway genes (III and IV) than non-pathway genes (I and II), and disease genes appeared in pathways involved in more diseases. The average degree of disease genes is larger than that of all genes; however, most disease genes are not hubs, resulting from low median degree. Values in italics are significantly increased in the corresponding rows.

| component | I | II | III | IV |
|---|---|---|---|---|
| number of human genes | 7104 | 3623 | 2708 | 3839 |
| median degree | 6 | 29 | 10 | 42 |
| average degree | 13.5075 | 43.0508 | 18.9524 | 71.1467 |
| number of disease genes | 698 | 528 | 790 | 1224 |
| proportion of disease genes (%) | 9.82 | 14.52 | *29.17* | *31.88* |
| median degree | 10 | 32 | 13 | 43 |
| average degree | 18.3840 | 46.8598 | 21.7747 | 76.0368 |
| average diseases involved | 1.4155 | 1.3314 | *2.0071* | *2.1047* |

## 3.4. Disease classification: in terms of the influence on pathways

The tacit assumption in network medicine is that the topological, functional and disease modules overlap, so that functional modules correspond to topological modules and a disease can be viewed as the breakdown of some particular functional modules [32,33].

**Table 4.** Module division obtained by fast unfolding algorithm, and the proportion of six types of pathway genes in 10 modules, respectively. The metabolism pathway genes are above-average proportion in modules 1, 7, 8. The genetic information processing pathway genes are above-average proportion in modules 3 and 4. Another four types of pathway genes are above-average proportion in modules 1, 2, 3 or 8. Values in italics are those greater than the proportion of genes in each column.

| module | 1 | 2 | 3 | 4 | 5 | 6 | 7 | 8 | 9 | 10 |
|---|---|---|---|---|---|---|---|---|---|---|
| number of genes | 3560 | 3269 | 2541 | 1820 | 1750 | 1639 | 1521 | 573 | 414 | 187 |
| proportion of genes | 0.206 | 0.189 | 0.147 | 0.105 | 0.101 | 0.095 | 0.088 | 0.033 | 0.024 | 0.011 |
| metabolism | *0.366* | 0.107 | 0.090 | 0.028 | 0.060 | 0.060 | *0.228* | *0.044* | 0.008 | 0.008 |
| genetic information processing | 0.114 | 0.054 | *0.240* | *0.345* | 0.056 | 0.076 | 0.084 | 0.004 | *0.025* | 0.002 |
| environmental information processing | *0.210* | *0.465* | 0.102 | 0.029 | 0.043 | 0.062 | 0.040 | 0.032 | 0.007 | 0.009 |
| cellular processes | 0.191 | *0.385* | 0.143 | 0.050 | 0.038 | 0.093 | 0.060 | 0.028 | 0.002 | 0.010 |
| organismal systems | *0.247* | *0.402* | 0.105 | 0.032 | 0.040 | 0.049 | 0.058 | *0.057* | 0.006 | 0.005 |
| human diseases | 0.174 | *0.334* | *0.208* | 0.077 | 0.039 | 0.058 | 0.056 | *0.043* | 0.004 | 0.006 |

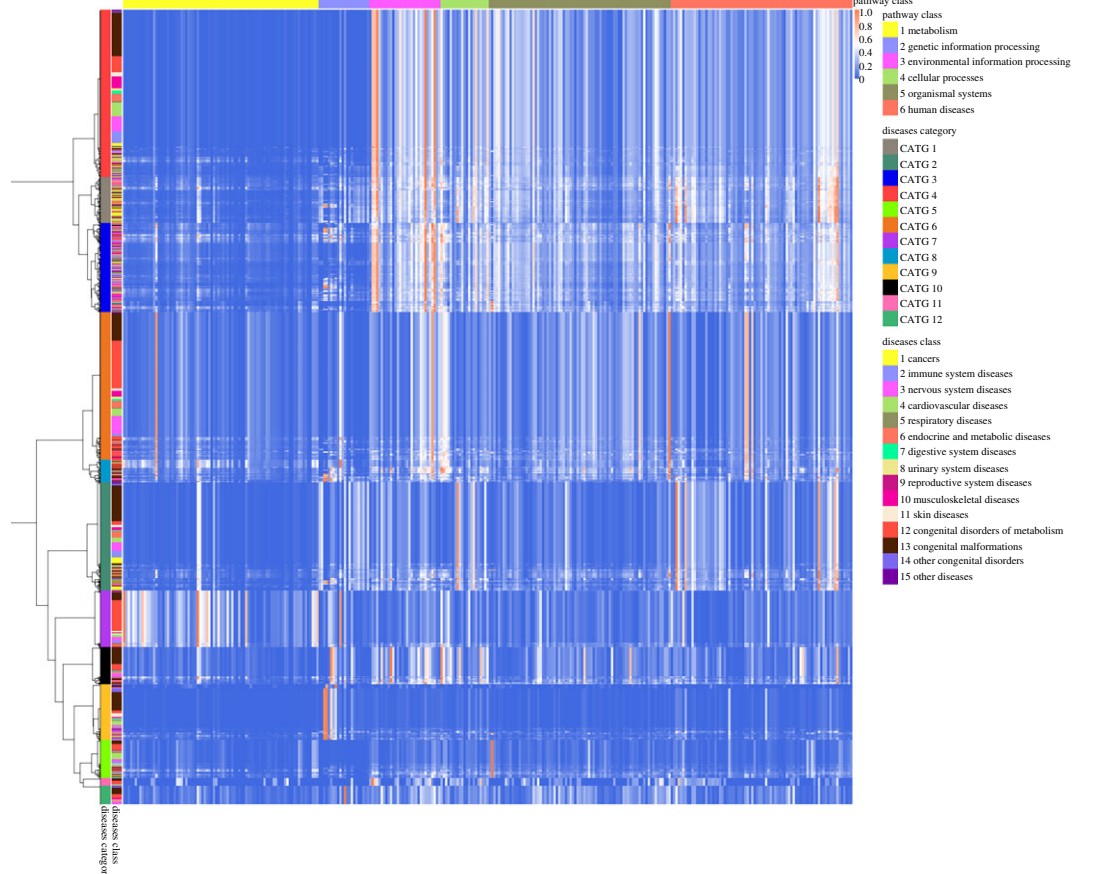

**Figure 4.** Hierarchical clustering of 1728 diseases by the intensity of effects on pathways yielded 12 categories (CATG 1, 2, . . ., 12). For each of 1728 diseases, we calculate and normalize a score vector $IS^N(D_k)$ corresponding to 317 pathways as the intensity of effects on pathways, and use Euclidean distance to measure the distance between two diseases.

A topological module represents a locally dense neighbourhood in a network, such that nodes have a higher tendency to link to nodes within the same local neighbourhood than to nodes outside it. A functional module represents the aggregation of nodes of similar or related function in the same network neighbourhood, where function captures the role of a gene in defining detectable phenotypes. Finally, a disease module represents a group of network components that together contribute to a cellular function and disruption of which results in a particular disease phenotype [32].

In this work, 10 topological modules are identified by network clustering algorithms; 317 pathways represent the function modules. For each of 1728 diseases, the collection of disease genes is regarded as a disease module. We perform different classification models of diseases for different interests. In the past, classification of diseases was mainly based on congenital and acquired factors as well as diseased tissues and organs, which did not help people to realize the influence on integrated pathway functions. Here, we consider disease genes, topological modules and functional pathways to assess the influence of diseases on pathways of human body (see details in Methods).

### 3.4.1. Classification by the intensity of effects on pathways

For each of 1728 diseases, we calculate and normalize a score vector $IS^N(D_k)$ corresponding to 317 pathways as the intensity of effects on pathways, and used Euclidean distance to measure the distance between two diseases. Hierarchical clustering of 1728 diseases by the intensity of effects on pathways yields 12 categories (see figure 4; detailed clustering results are available in electronic supplementary material), as the curve of Difference$^P$ changes and get the minimum when 1716 merges are conducted (figure 6a). The distance matrix between 1728 diseases arranged in the clustering order corresponding to the dendrogram is illustrated in figure 6c. Each of 12 categories is a fairly heterogeneous mix of several types of diseases, which suggests that diseases of different pathological classifications may have similar intensity of effects on pathways. Cancers mainly are grouped in CATG 1 as well as CATG 2, CATG 3,

CATG 4. Most cancers in CATG 2 and CATG 4 are of haematopoietic and lymphoid tissues, and of soft tissues and bone. Only three cancers (anaplastic large-cell lymphoma, nasopharyngeal cancer and neuroblastoma) are grouped in CATG 3. In CATG 1, many other diseases, such as fanconi anaemia and type II diabetes mellitus, are grouped together with many cancers, showing that these diseases affect human pathways similarly and seriously as cancers. Fanconi anemia (FA) is a genetic disorder that is characterized by bone marrow failure, developmental abnormalities and predisposition to cancer. Monoallelic inactivation of some FA genes, such as FA complementation group D1 (FANCD1, also known as the breast and ovarian cancer susceptibility gene BRCA2), leads to adult-onset cancer predisposition but does not cause FA, and somatic mutations in FA genes occur in cancers in the general population. Studies of FA have revealed opportunities to develop rational therapeutics for this genetic disease and for malignancies that acquire somatic mutations within the FA pathway [46,47]. Type II diabetes mellitus is a long-term metabolic disorder, and the relationship between type II diabetes mellitus and cancers has always been researched. Several studies have suggested that diabetes mellitus may alter the risk of developing a variety of cancers, and the associations are biologically plausible [48]. Coughlin *et al.* [48] suggest that diabetes is an independent predictor of mortality from cancer of the colon, pancreas, female breast, and, in men, of the liver and bladder. Yang *et al.* [49] show that chronic insulin therapy significantly increases the risk of colorectal cancer among type II diabetes mellitus patients. Huxley *et al.* [50] conduct a meta-analysis and the results supported a modest causal association between type II diabetes and pancreatic cancer.

Diseases in CATG 1, CATG 3, CATG 4 mainly affect such functions of regulating cell proliferation, survival, growth, migration, differentiation, adhesion by affecting ras signalling pathway, rap1 signalling pathway, MAPK signalling pathway, Jak-STAT signalling pathway and PI3K-Akt signalling pathway. Diseases in CATG 6, most of which are metabolic diseases, mainly affect oxidative phosphorylation pathway, neuroactive ligand–receptor interaction pathway, thermogenesis pathway, Alzheimer's disease pathway, Parkinson's disease pathway, Huntington's disease pathway. However, Alzheimer's disease and Huntington's disease are in CATG 4 and Parkinson's disease is in CATG 3. Compared with CATG 6, diseases in CATG 8 also affect many genetic information processing pathways and environmental information processing pathways. Diseases in CATG 2 are associated with virus infection and carcinogenesis, as they mainly affect cell cycle pathway, cellular senescence pathway, transcriptional misregulation in cancer pathway, viral carcinogenesis pathway, HTLV-I infection pathway, Epstein–Barr virus infection pathway, human papillomavirus infection pathway. Diseases in CATG 7 mainly affect metabolism pathways and genetic information processing pathways, such as glycolysis/gluconeogenesis pathway, purine metabolism pathway, protein processing in endoplasmic reticulum pathway, ubiquitin-mediated proteolysis pathway. Moreover, endocytosis pathway is affected so much. Diseases in CATG 10 mainly affect RNA transport pathway, mRNA surveillance pathway, Hippo signalling pathway, endocytosis pathway, oocyte meiosis pathway, tight junction pathway, adrenergic signalling in cardiomyocytes pathway, dopaminergic synapse pathway, human papillomavirus infection pathway. Diseases in CATG 9 mainly affect genetic information processing pathways including spliceosome pathway, ribosome pathway and RNA transport pathway. Diseases in CATG 5 mainly affect complement and coagulation cascades pathway. Diseases in CATG 11 mainly affect ras signalling pathway, tight junction pathway, microRNAs in cancer pathway. Diseases in CATG 12 mainly affect ubiquitin-mediated proteolysis pathway.

### 3.4.2. Classification by the scope of effects on pathways

For each of 1728 diseases, we use a binary vector $IS^B(D_k)$ corresponding to 317 pathways as the scope of effects on pathways, and use Manhattan distance to measure the distance between two diseases. Hierarchical clustering of 1728 diseases by the scope of effects on pathways yields 14 categories (see figure 5; detailed clustering results are available in electronic supplementary material), as the curve of Difference$^P$ changes and get the minimum when 1714 merges are conducted (figure 6*b*). The distance matrix between 1728 diseases arranged in the clustering order corresponding to the dendrogram is illustrated in figure 6*d*.

Diseases in CATG I mainly affect nucleotide metabolism pathways, signal transduction pathways, cellular community pathways, cell motility pathways, development pathways, cancers pathways, viral infectious diseases pathways. Diseases in CATG II mainly affect nucleotide metabolism pathways, replication and repair pathways, cancers: specific types pathways, viral infectious diseases pathways. Diseases in CATG III mainly affect signal transduction pathways, signalling molecules and interaction pathways, cellular community pathways, cell motility pathways, immune system pathways, development pathways, cancers

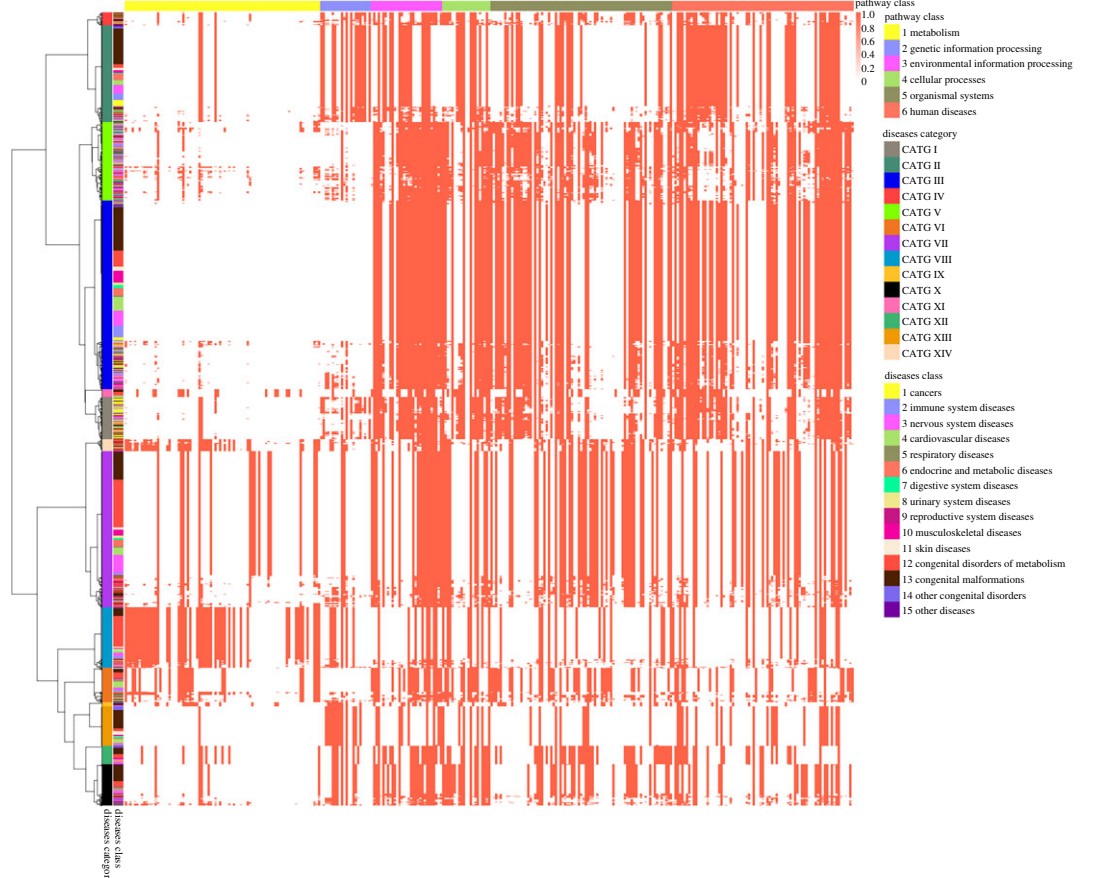

**Figure 5.** Hierarchical clustering of 1728 diseases by the scope of effects on pathways yielded 14 categories (CATG I, II, . . ., XIV). For each of 1728 diseases, we use a binary vector $IS^B(D_k)$ corresponding to 317 pathways as the scope of effects on pathways, and use Manhattan distance to measure the distance between two diseases.

pathways. Diseases in CATG IV mainly affect nucleotide metabolism pathways, development pathways, viral infectious diseases pathways. Diseases in CATG V mainly affect signalling molecules and interaction pathways, cellular community pathways, cell motility pathways, circulatory system pathways, development pathways, viral infectious diseases pathways. Diseases in CATG VI mainly affect xenobiotics biodegradation and metabolism pathways, signalling molecules and interaction pathways, cell motility pathways. Diseases in CATG VII mainly affect membrane transport pathways, signalling molecules and interaction pathways. Diseases in CATG VIII mainly affect carbohydrate metabolism pathways, nucleotide metabolism pathways, metabolism of terpenoids and polyketides pathways, xenobiotics biodegradation and metabolism pathways, cell motility pathways, ageing pathways. Diseases in CATG IX mainly affect translation pathways. Diseases in CATG X mainly affect nucleotide metabolism pathways, transport and catabolism pathways, cell motility pathways. Diseases in CATG XI mainly affect membrane transport pathways, cellular community pathways, cell motility pathways, development pathways, ageing pathways, cancers pathways. Diseases in CATG XII mainly affect nucleotide metabolism pathways, cellular community pathways, cell motility pathways, circulatory system pathways, development pathways, ageing pathways, environmental adaptation pathways, cancers pathways. Diseases in CATG XIII mainly affect translation pathways, cell motility pathways, circulatory system pathways. Diseases in CATG XIV mainly affect nucleotide metabolism pathways, xenobiotics biodegradation and metabolism pathways, membrane transport pathways, cell motility pathways.

### 3.4.3. Associations and differences between the two classifications

In this paper, we integrate disease genes, topological modules and functional pathways to assess the influence of diseases on pathways of human body. When it comes to associations between the two classifications, the first is that both of them are based on the impact score, which is the basic metric in this paper (more details can be found in §2.4). The second is the criteria for classifications (more

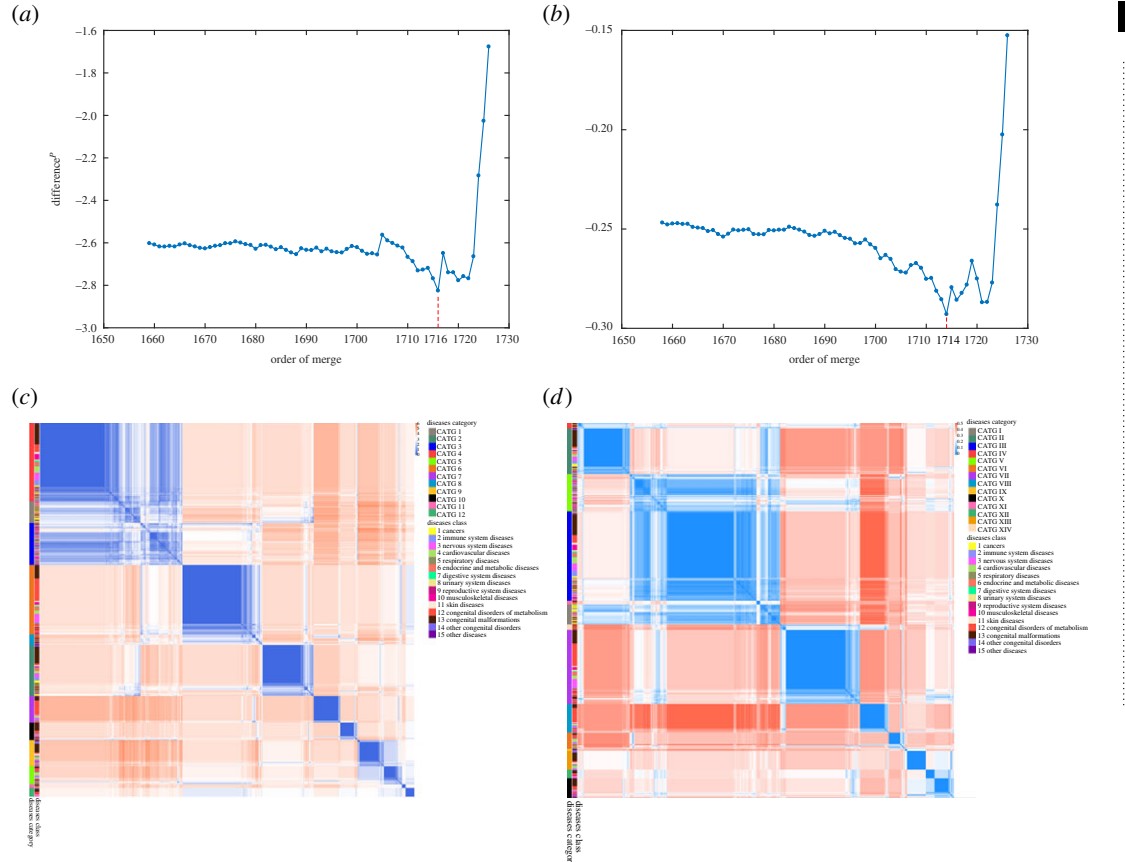

**Figure 6.** Corresponding to the dendrogram in figure 4, partitions $\{P_{1659}, P_{1660}, \ldots, P_{1727}\}$ are considered in which every disease is merged at least once. Difference$^P$ gets the minimum when 1716 merges are conducted and revealed 12 categories (*a*), and the distance matrix between 1728 diseases arranged in the clustering order is illustrated in (*c*). Corresponding to the dendrogram in figure 5, partitions $\{P_{1658}, P_{1659}, \ldots, P_{1727}\}$ are considered. Difference$^P$ gets the minimum when 1714 merges are conducted and revealed 14 categories (*b*), and the distance matrix between 1728 diseases arranged in the clustering order is illustrated in (*d*).

details can be found in §2.5 and figure 6). The third is that most disease pairs are grouped together in both of the two classifications (illustrated in grey numbers in table 5, as the maximum either in the row or in the column); however, many of them are not in the same group in KEGG DISEASE database (figure 7*a*).

For example, in the middle of figure 7*a*, it shows that many cancers have common disease genes and are also classified into one group in our classification. What is more, the giant cell tumour of bone (#699, classified as a musculoskeletal disease in KEGG DISEASE, see the electronic supplementary material) is closely connected with these cancer diseases. Giant cell tumour of the bone is a relatively uncommon tumour of the bone. Malignancy in giant cell tumour is uncommon; however, if malignant degeneration does occur, it is likely to metastasize to the lungs [51]. So it is reasonable to group giant cell tumour of bone with many cancers including non-small cell lung cancer (#19) and small cell lung cancer (#20).

In the top of figure 7*a*, many allergies and autoimmune diseases including asthma (#85), systemic lupus erythematosus (#86), Hashimoto thyroiditis (#87), Graves' disease (#88), dermatitis herpetiformis (#99), systemic sclerosis (#102 in immune systems diseases, #805 in skin diseases), Sjogren syndrome (#103 in immune systems diseases, #806 in skin diseases), are grouped together with dilated cardiomyopathy (#389 in cardiovascular diseases), celiac disease (#637 in digestive system diseases), etc. One of the reasons is that HLA class II alpha chain paralogues (HLA-DQA1, HLA-DQB1, HLA-DRB1) are the common disease genes which play a central role in the immune system by presenting peptides derived from extracellular proteins.

In the right of figure 7*a*, congenital muscular dystrophies (#711 in musculoskeletal diseases), muscular dystrophy–dystroglycanopathy type A (#717 in musculoskeletal diseases, #950 in congenital disorders of metabolism, #1197 in congenital malformations), muscular dystrophy–dystroglycanopathy type B (#718

**Table 5.** The overlap of diseases between the two classifications. The numbers in grey are the maximum either in the row or in the column, representing the correspondence between the two classifications. Other positive numbers imply that minority of diseases are grouped differently in the two classifications.

| CATG | 1 | 2 | 3 | 4 | 5 | 6 | 7 | 8 | 9 | 10 | 11 | 12 |
|------|---|---|---|---|---|---|---|---|---|----|----|----|
| I | 83 | 9 | | | | | | | | | | |
| II | | 207 | | | | | | | 4 | | | |
| III | 4 | | 44 | 363 | | | | | | | | |
| IV | 5 | 19 | | | | | | 4 | | | | |
| V | 8 | | 149 | 1 | 10 | | | 1 | 1 | 1 | | |
| VI | | | | | 71 | | | | 4 | | | |
| VII | | | 1 | | 1 | 314 | | 22 | 2 | | | |
| VIII | | | | | | | 123 | 2 | 5 | | 2 | |
| IX | | | | | | | | | 9 | | | |
| X | | | | | | | | 2 | 10 | 78 | | |
| XI | | | | | | | | | | | 17 | |
| XII | | | | | | | | | | | | 40 |
| XIII | | | | | | | | | 86 | | | |
| XIV | | | | | 1 | 7 | | 18 | | | | |

in musculoskeletal diseases, #951 in congenital disorders of metabolism), muscular dystrophy–dystroglycanopathy type C (#719), Fukuyama congenital muscular dystrophy (#720), congenital muscular dystrophy type 1C (#721 in musculoskeletal diseases, #952 in congenital disorders of metabolism), congenital muscular dystrophy type 1D (#722 in musculoskeletal diseases) are closely connected with each other. The fact that one disease may be classified as more than one category in KEGG DISEASE database motivates us to find a method to understand the relationships between different categories of disease from the genetic level.

When it comes to differences between the two classifications, the first is that they emphasize the impact of disease on pathways from different perspectives. To classify diseases by the intensity of effects on pathways, we use the normalized vector $IS^N(D_k)$ to measure the difference in intensity between the pathways. The score of the most affected pathway is 1, and the score of the least affected pathway is 0. To classify diseases by the scope of effects on pathways, we use the binary vector $IS^B(D_k)$ to mark pathways with a score exceeding the average. The second is the distance of diseases. In the first classification, the distance of diseases $D_i$ and $D_j$ is defined as the Euclidean distance of $IS^N(D_i)$ and $IS^N(D_j)$. In the second classification, the distance of diseases $D_i$ and $D_j$ is defined as the Manhattan distance of $IS^B(D_i)$ and $IS^B(D_j)$. Note that for two binary vectors, the following equation holds: $\text{distance}_{\text{manhattan}} \times \text{dimension}_{\text{vector}} = (\text{distance}_{\text{euclidean}})^2$. The third is that there exist some disease pairs that are grouped differently in the two classifications (figure 7b,c).

If the diseases are only grouped together in the first classification, it means that the impact on function by these diseases is concentrated in certain pathways but each disease uniquely affects several other pathways. The pair of type II diabetes mellitus (#536) and basal cell carcinoma (#23) is an example. Besides, hypertrophic cardiomyopathy (#387) and dilated cardiomyopathy (#389), retinitis pigmentosa (#284) and leber congenital amaurosis (#306), deafness, autosomal dominant (#346) and deafness, autosomal recessive (#347) are other disease pairs that are only grouped together in the first classification (figure 7b).

If the diseases are only grouped together in the second classification, it means that these diseases may affect similar pathways, but with different intensities. For example, laryngeal cancer (#6), fallopian tube cancer (#44), chronic myeloid leukaemia (#77), myelodysplastic syndrome (#506) are four of nine diseases that are grouped together in CATG 2 and CATG I (figure 7c and table 5), which implies they are special. In the treatment of these diseases, it should be done to pay attention to the pathways they affect; however, the focuses are different. Additionally, glycogen storage diseases (#835) and hepatic glycogen storage disease (#836), progressive external ophthalmoplegia (#330, #1145) and mitochondrial DNA depletion syndrome (#1138) are other disease pairs that are only grouped together in the second classification (figure 7c).

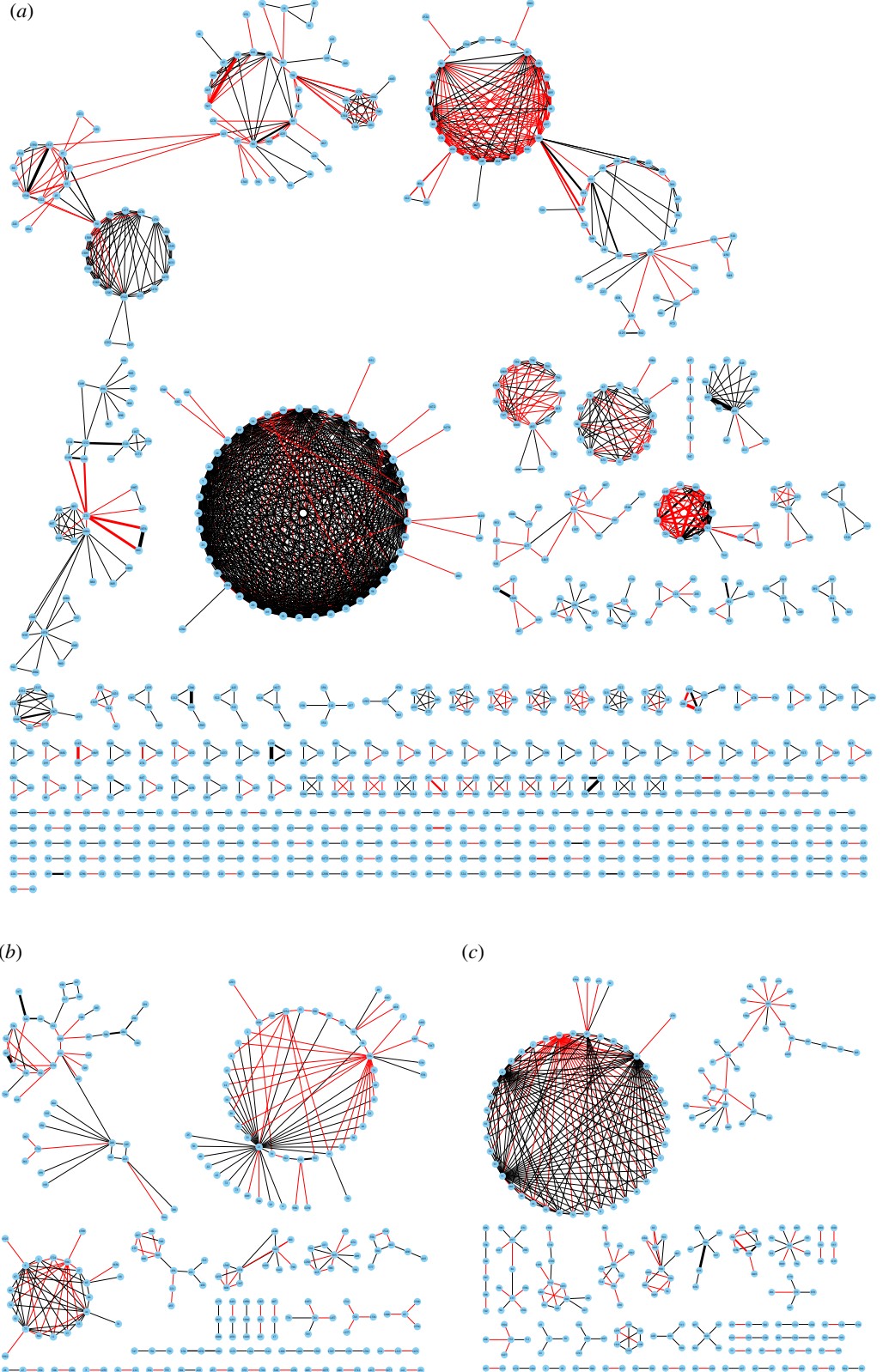

**Figure 7.** Illustration of diseases networks. The necessary condition for the link between two diseases is that they have at least one common disease gene. (*a*) The disease network consists of disease pairs that are grouped together in both classifications. (*b*) The disease network consists of disease pairs that are grouped together in only the first classification. (*c*) The disease network consists of disease pairs that are grouped together in only the second classification. Black links indicate that the corresponding two diseases are grouped together in the KEGG DISEASE database, while the red links indicate the opposite. The width of the edge is proportional to the number of common disease genes of the two diseases. The disease name and H number in KEGG DISEASE corresponding to its number in this paper can be found in the electronic supplementary material.

# 4. Discussion

In 2016, the precision medicine initiative was announced to help enable a new era of personalized care through cooperative efforts by researchers, clinicians and patients. Since then, researchers are trying to figure out what set of technologies and disciplines would afford the highest level of efficacy in the development of precision medicine [52]. The study of genomics and other molecular analyses at the omics level are rapidly growing fields that have the potential to have a profound impact upon medical practice. Many good studies focus on underlying molecular mechanisms of diseases [26,29,53], providing predictive, prognostic, diagnostic and surrogate markers of diverse disease states [54,55], classifying patient samples based on molecular data [27,28].

Most of the successful studies building on these new approaches have focused on a single disease or a class of diseases to gain a better understanding. Recent progress in genetics and genomics has led to an appreciation of the effects of gene mutations in virtually all disorders and provides the opportunity to study human diseases all at once rather than one at a time. Under the key hypothesis that a disease phenotype is rarely a consequence of an abnormality in a single effector gene product but reflects various pathobiological processes that interact in a complex network [32], the network-based approaches offer the possibility of discerning general patterns and correlations of human disease not readily apparent from the study of individual disorders [22]. To address the fundamental challenge of modern biomedical research that understanding how diseases that are similar on the phenotypic level are similar on the molecular level [4], our focus is on classifying diseases based on pathways impacted and supporting combination therapy between diseases as evidence. The distances between diseases based on the human gene network and pathways are defined to evaluate the similarity of diseases, even the similarity of gene-targeted therapies.

It is considered to be right that disease genes affect pathway functions through topology modules [32,33]. Therefore, we derive the influence on pathways by diseases through calculating inner product of the following two vectors. The first vector is used to measure the propagation efficiency of specific disease signals in each of the modules. Each component of the vector is the summation of the closeness centrality of disease genes within the corresponding module. Mathematically, the technique of network propagation is simplifying and unifying. It is a powerful data transformation method of broad utility in genetic research, since it greatly improves the power of genetic association, providing a universal amplifier for genetic analysis [56]. The second vector is a measure of relevance between the module and pathways. Each component of the vector is the jaccard similarity coefficient of the module and corresponding pathway, indicating the extent of overlap of two large gene sets. This measure is selected mainly because it has proven to do well in comparing two sets of nodes when considering the difference of the size of the two sets [57]. Moreover, it provides an intuitive way to characterize the set similarity.

Hierarchical clustering has been the dominant approach to constructing classification schemes, and much early work on hierarchical clustering was in the field of biological taxonomy from the 1950s and more so from the 1960s onwards [58]. The dendrogram expresses many of the proximity and classificatory relationships in a body of data. To answer the question 'how many groups are there?', we defined a parameter to measure the difference of average distances of diseases within groups to average distances of distances between groups, playing the same role as modularity in module partition.

In total, 1728 diseases collected from KEGG are divided into 12 categories by the intensity of effects on pathways, and are identified as 14 categories by the scope of effects on pathways. Each category is a mix of several types of diseases identified based on congenital and acquired factors as well as diseased tissues and organs. The number of disease categories that contain cancers is the smallest among 15 types of diseases, which suggests the similarity of cancer diseases in terms of having a great impact on pathways, because almost all cancers involve multiple genes. As for monogenic diseases, the disease module is regarded as the disease gene, such that the distance between two diseases is zero when the disease genes belong to the same topological module, resulting in the situation that the two diseases are grouped together. Otherwise, the two diseases will be divided into different categories. The number of disease categories will be larger than the number of topological modules generally. Our results imply that the human gene network gives a perspective of disease classifications.

The method for deriving the results of this paper is based on the topological structure of gene interaction network especially the LCC and module division results. After the completion of the Human Genome Project, the number of new genes discovered in the future should be very small. However, it is unavoidable that the topological structure of gene interaction network will be different because of the exploration of new gene–gene interactions. We take the published time of the literature as the time of discovery of interactions, based on which we obtain the LCC of the human gene

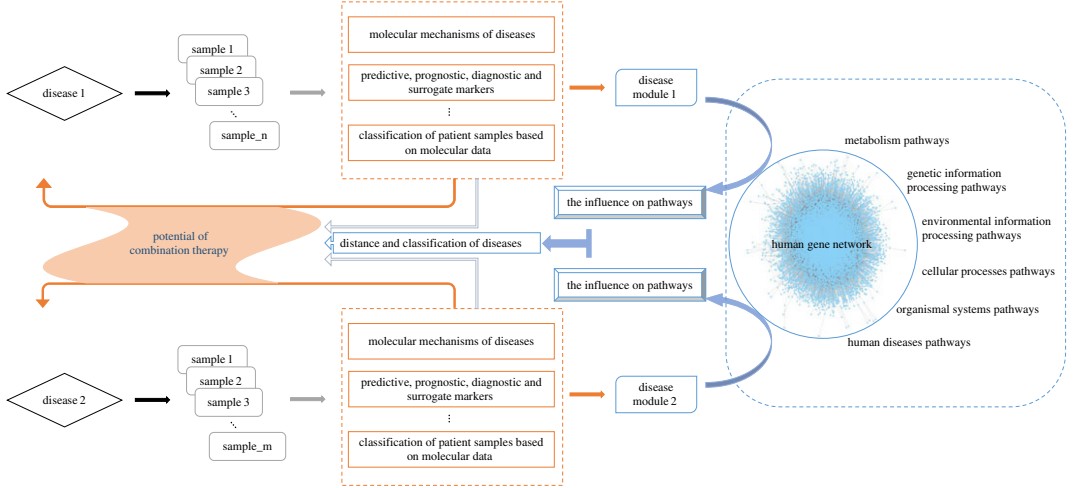

**Figure 8.** Molecular data of patient samples for concerned diseases yield good results in many great researches, which have been providing valuable guidance to precision medicine from many aspects, such as providing predictive, prognostic, diagnostic and surrogate markers of diverse disease states, informing on underlying molecular mechanisms of diseases, allowing for classification of patients based on molecular data, etc. Our work (the blue part) is an extension on their basis, firstly, because the data we need, for example disease genes, are identified and summarized in previous researches. Secondly, the direct application of genetic information can be considered as first order, while the human gene network is an integration of genetic information, which is high order. Thirdly, our classifications of diseases are at the system level, designed to provide novel insights for clinical practice at the sample level, for example, a repositioning of our understanding of diseases and exploration of the potential of combination therapy.

network at the end of each year (from 2003 to 2017). The number of genes in the LCC increases; however, the number of modules does not change much (more details are available in the electronic supplementary material). This fact may be due to properties of scale-free networks that most interactions newly discovered connect a hub gene and an isolated gene. Although the human gene network will inevitably undergo local changes with the discovery of new genes and gene–gene interactions, the overall organization and layout of the network will not be changed significantly, because it is unchanged that hub nodes play a major role in the modules. As gene–gene interaction relations are constantly explored, gene interaction networks are evolving. To describe how much a gene has been studied, an approach called gene saturation which is based on a logistic model for each gene has been proposed recently [59]. This approach may provide some guidances for experimental researchers to choose their research object and discover new gene–gene interactions efficiently.

Disease classification is a progression towards precision medicine with the need for precise patient characterization, currently based on clinical phenotypes but in future augmented by laboratory-based tests [60]. As illustrated in figure 8, researchers of hospitals and institutions obtain molecular data of patient samples for diseases of concern. Their great studies have been providing valuable guidance to precision medicine from many aspects, such as providing predictive, prognostic, diagnostic and surrogate markers of diverse disease states, informing on underlying molecular mechanisms of diseases, allowing for classification of patients based on molecular data, etc. Our work (the blue part) is an extension on their basis, firstly, because the data we need, for example disease genes, are identified and summarized in previous researches. Secondly, the direct application of genetic information can be considered as first order, while the human gene network is an integration of genetic information, which is high order. Thirdly, our classifications of diseases are at the system level, designed to provide novel insights for clinical practice at the sample level, for example, a repositioning of our understanding of diseases and exploration of the potential of combination therapy. Also, our results of disease classifications may complement each other with the classification of complications, as a clear definition of complications is essential in medicine, mainly aiming to improve quality in patients' care. The lack of method for uniform reporting of complications both in terms of definition and grading prompted the authors to propose a classification system of complications based on combining outcome and severity of sequelae [61]. The integration of such work will play a role in guiding combination therapy. Moreover, the enormous complexity of common diseases and the resulting problems, such as the fact that many patients do not respond to treatment and the increasing costs of drugs and drug development, provide

strong motivation for new and complementary strategies for research and clinical practice [62]. The network-based approaches classifying diseases based on pathways impacted have the potential to substantially enable the elaboration of a network based view of drug discovery and reposition (the application of known drugs to new indications), which are challenging issues in pharmaceutical science [63,64].

Prior to clinical implementation, major challenges must be addressed from a clinician's perspective, including understanding how network approach based genomic science is generated and linked to patient-oriented science, which is the framework for evaluating genomic studies, as an evidence base for providing effective precision medicine to patients in the future [65]. The diseases studied in our work are far more than other researches, not limited to cancers or certain diseases. This fact may lead to our results being extensive; however, it may guide people to pay attention to some unexpected points.

A comprehensive description of the associations between pathways and diseases requires identification of not only multiple pathways associated with a specific disease but also pathways associated with multiple diseases. In 9125 tumour samples, Sanchezvega *et al.* [29] point out significant representation of individual and co-occurring actionable alterations in 10 signalling pathways. Meanwhile, we find that most of the 10 signalling pathways are subject to significant influence by diseases, including Hippo signalling pathway, PI3K-Akt signalling pathway, Notch signalling pathway, p53 signalling pathway, cell cycle pathway, Ras signalling pathway, TGF-beta signalling pathway, Wnt signalling pathway. The current understanding of the Hippo pathway, for example, has been reviewed with an emphasis on the effects of this pathway on basic biology and human diseases, including cancers, immunity and cardiovascular diseases [66]. Another review shows that individuals with RASopathies share many overlapping characteristics, including cardiac malformations, short stature, neurocognitive impairment, craniofacial dysmorphy, cutaneous, musculoskeletal, and ocular abnormalities, hypotonia and a predisposition to developing cancer [67]. These results suggest resemblance between cancer diseases and non-cancer diseases, and implies opportunities for targeted and combination therapies. Integrative analysis methods have been proposed to improve power and reproducibility for identifying genes and prognosis markers associated with multiple cancers, which may lead to discovery of novel therapeutic targets for cancer therapies [68,69]. In addition, it estimates the effect (either positive or negative) of the same gene in different diseases, which helps predict possible side effects of a drug. Supported by our classification of diseases, similar applications can be implemented between cancer diseases and non-cancer diseases to discover biomarker targets and improve drug development for multiple diseases that are classified in the same category.

Data accessibility. The data calculated in results are provided as electronic supplementary material.

Authors' contributions. Z.M., B.G. and Z.Z. conceived the idea for this study. Z.M. performed the theoretical and computational analyses. B.G., Z.Y. and J.L. co-supervised the analyses and gave suggestions. Z.M. drafted the manuscript. All authors have read and approved the final manuscript.

Competing interests. We declare we have no competing interests.

Funding. This work was supported by the Major Program of National Natural Science Foundation of China (11290141), National Natural Science Foundation of China (11401017, 11571028 and 11671025), Fundamental Research Funds for the Central Universities (BUAA), and Fundamental Research of Civil Aircraft no. MJ-F-2012-04.

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
