## [Reviewer comments · Royal Society Open Science]

Review History

RSOS-190214.R0 (Original submission)

Review form: Reviewer 1

Is the manuscript scientifically sound in its present form?

Yes

Are the interpretations and conclusions justified by the results?

Yes

Is the language acceptable?

Yes

Is it clear how to access all supporting data?

Not Applicable

Do you have any ethical concerns with this paper?

No

Have you any concerns about statistical analyses in this paper?

Yes

Recommendation?

Accept with minor revision (please list in comments)

Comments to the Author(s)

The manuscript is dedicated to the diseases classification, which is the foundation for achieving precision medicine. Considering that diseases genes affect pathway functions through topology modules, the authors collected data from NCBI and KEGG and carried out diseases classification models from the perspective of gene network modules and pathways. In addition, the authors were trying to perform different classification models of diseases taking account of intensity and scope of the impact of the disease on pathways. The study is solid and the results reported are interesting. After close evaluation the manuscript I have some comments and questions:

1 The authors summarized many databases of different institutions and organizations in the introduction, why not combine all databases but only used data from NCBI and KEGG?

2 It will make sense if the authors appropriately pointed out good examples of diseases that are grouped together in Figure 4 and Figure 5.

3 Associations and differences between the two classifications should be described in more details.

4 How to assess the impact of discovering new genes in the future on current results?

5 Line 32 in page 5, "play a role in develop the disease" should be "plays a role in developing the disease".

6 Line 37 in page 5, "Selecting this measure mainly because" should be "This measure is selected mainly because".

Review form: Reviewer 2**Is the manuscript scientifically sound in its present form?**

No

Are the interpretations and conclusions justified by the results?

No

Is the language acceptable?

Yes

Is it clear how to access all supporting data?

Not Applicable

Do you have any ethical concerns with this paper?

No

Have you any concerns about statistical analyses in this paper?

I do not feel qualified to assess the statistics

Recommendation?

Major revision is needed (please make suggestions in comments)

Comments to the Author(s)

This paper proposed several measures to quantify the effect of diseases on gene pathways and functional networks and to quantify the distance between diseases and used these measures to perform hierarchical clustering of diseases. The authors essentially performed unsupervised learning based on gene networks based on diseases information. The authors may explain in detail how the analysis results especially those based on the disease distance can be useful in precision medicine (as claimed by the authors), in comparison to many developed methods in the literature for predicting disease based on gene information, it is unclear

The authors should have explained the scientific motivations for defining those quantities and for performing clustering of diseases.

Decision letter (RSOS-190214.R0)

29-Apr-2019

Dear Mr Mi,

The editors assigned to your paper ("Diseases classification via gene network integrating modules and pathways.") have now received comments from reviewers.

While the reviewers and editor are positive about publication of your paper, they raise some substantive issues, particularly with regard to the impact of your study on practice in the clinic. Articulating this impact and the potential future wider relevance of your studies and conclusions in genomic medicine and clinical practice will be vital. It will be important for you to revise your paper in accordance with the referee and Associate Editor suggestions which can be found below (not including confidential reports to the Editor). Please note this decision does not guarantee eventual acceptance. Please further note that you must seek the advice of a language polishing service (<https://royalsociety.org/journals/authors/language-polishing/>) or a colleague who is a native speaker of English before resubmitting.

Please submit a copy of your revised paper before 22-May-2019. Please note that the revision deadline will expire at 00.00am on this date. If we do not hear from you within this time then it will be assumed that the paper has been withdrawn. In exceptional circumstances, extensions may be possible if agreed with the Editorial Office in advance. We do not allow multiple rounds of revision so we urge you to make every effort to fully address all of the comments at this stage. If deemed necessary by the Editors, your manuscript will be sent back to one or more of the original reviewers for assessment. If the original reviewers are not available, we may invite new reviewers.

When submitting your revised manuscript, you must respond to the comments made by the referees and upload a file "Response to Referees" in "Section 6 - File Upload". Please use this to document how you have responded to the comments, and the adjustments you have made. In

order to expedite the processing of the revised manuscript, please be as specific as possible in your response.

- Data accessibility

If you wish to submit your supporting data or code to Dryad (<http://datadryad.org/>), or modify your current submission to dryad, please use the following link:
<http://datadryad.org/submit?journalID=RSOS&manu=RSOS-190214>

- Competing interests

- Authors' contributions

- Acknowledgements

- Funding statement

on behalf of Professor Joris Veltman (Associate Editor) and Steve Brown (Subject Editor)
openscience@royalsociety.org

Associate Editor's comments (Professor Joris Veltman):

The manuscript by Zhilong et al. provides important new approaches to disease classification. While the results section is extensive, I do miss a thorough discussion of the use of this in the clinic. A better motivation of the choice for the gene network based approach needs to be provided as well as a thorough discussion on the potential impact of applying this approach in disease classification.

Also, it is important to have the paper checked for spelling mistakes.

Comments to Author:

Reviewers' Comments to Author:

Reviewer: 1

Comments to the Author(s)

The manuscript is dedicated to the diseases classification, which is the foundation for achieving precision medicine. Considering that diseases genes affect pathway functions through topology modules, the authors collected data from NCBI and KEGG and carried out diseases classification models from the perspective of gene network modules and pathways. In addition, the authors were trying to perform different classification models of diseases taking account of intensity and scope of the impact of the disease on pathways. The study is solid and the results reported are interesting. After close evaluation the manuscript I have some comments and questions:

1 The authors summarized many databases of different institutions and organizations in the introduction, why not combine all databases but only used data from NCBI and KEGG?

2 It will make sense if the authors appropriately pointed out good examples of diseases that are grouped together in Figure 4 and Figure 5.

3 Associations and differences between the two classifications should be described in more details.

4 How to assess the impact of discovering new genes in the future on current results?

5 Line 32 in page 5, "play a role in develop the disease" should be "plays a role in developing the disease".

6 Line 37 in page 5, "Selecting this measure mainly because" should be "This measure is selected mainly because".

Reviewer: 2

Comments to the Author(s)

This paper proposed several measures to quantify the effect of diseases on gene pathways and functional networks and to quantify the distance between diseases and used these measures to perform hierarchical clustering of diseases. The authors essentially performed unsupervised learning based on gene networks based on diseases information. The authors may explain in detail how the analysis results especially those based on the disease distance can be useful in precision medicine (as claimed by the authors), in comparison to many developed methods in the literature for predicting disease based on gene information, it is unclear

The authors should have explained the scientific motivations for defining those quantities and for performing clustering of diseases.

Author's Response to Decision Letter for (RSOS-190214.R0)

See Appendix A.

Decision letter (RSOS-190214.R1)

04-Jun-2019

Dear Mr Mi,

I am pleased to inform you that your manuscript entitled "Diseases classification via gene network integrating modules and pathways." is now accepted for publication in Royal Society Open Science.

on behalf of Professor Joris Veltman (Associate Editor) and Steve Brown (Subject Editor)
openscience@royalsociety.org

Associate Editor Comments to Author (Professor Joris Veltman):

The authors have adapted the manuscript extensively based on the reviewers as well as my recommendations. This has greatly improved the manuscript and I see no further issues that need to be addressed.

Appendix A

Response to the comments on manuscript ID RSOS-190214 “Diseases classification via gene network integrating modules and pathways.”

Associate Editor's comments (Professor Joris Veltman):

The manuscript by Zhilong et al. provides important new approaches to disease classification. *While the results section is extensive, I do miss a thorough discussion of the use of this in the clinic. A better motivation of the choice for the gene network based approach needs to be provided as well as a thorough discussion on the potential impact of applying this approach in disease classification.*

Response:

Thank you very much for your suggestion. We have discussed this question in the section ‘Discussion’, as following:

- 1) The motivation of the choice for the gene network based approach. (2nd paragraph in Discussion)

Most of the successful studies building on these new approaches have focused on a single disease or a class of diseases to gain a better understanding. Recent progress in genetics and genomics has led to an appreciation of the effects of gene mutations in virtually all disorders and provides the opportunity to study human diseases all at once rather than one at a time. Under the key hypothesis that a disease phenotype is rarely a consequence of an abnormality in a single effector gene product but reflects various pathobiological processes that interact in a complex network[32], the network-based approaches offer the possibility of discerning general patterns and correlations of human disease not readily apparent from the study of individual disorders[22]. To address the fundamental challenge of modern biomedical research that understanding how diseases that are similar on the phenotypic level are similar on the molecular level[4], our focus is on classifying diseases based on pathways impacted and supporting combination therapy between diseases as evidence. The distances between diseases based on the human gene network and pathways are defined to evaluate the similarity of diseases, even the similarity of gene-targeted therapies.

- 2) The potential impact of applying this approach in disease classification. (7th to 9th paragraphs in Discussion)

Disease classification is a progression towards precision medicine with the need for precise patient characterization, currently based on clinical phenotypes but in future augmented by laboratory-based tests[60]. As illustrated in Fig.8, researchers of hospitals and institutions obtain molecular data of patient samples for diseases of concern. Their great studies have been providing valuable guidance to precision medicine from many aspects, such as providing predictive, prognostic, diagnostic, and surrogate markers of diverse disease states, informing on underlying molecular mechanisms of diseases, allowing for classification of patients based on molecular data etc. Our work (the blue part) is an extension on their basis, firstly, because the data we need, for example, disease genes are identified and summarized in previous researches. Secondly, the direct application of genetic information can be considered as first-order, while the human gene network is an integration of genetic information, which is high-order. Thirdly, our classifications of diseases are at a system level, designed to provide novel insights for the clinic practice at the

sample level, for example a repositioning of our understanding of diseases and exploration the potential of combination therapy. Also, our results of diseases classifications may complement each other with the classification of complications, as a clear definition of complications is essential in medicine, mainly aiming to improve quality in patients' care. The lack of method for uniform reporting of complications both in terms of definition and grading prompted the authors to propose a classification system of complications based on combining outcome and severity of sequelae[61]. The integration of such work will play a role in guiding combination therapy. Moreover, the enormous complexity of common diseases and the resulting problems, such as the fact that many patients do not respond to treatment and the increasing costs of drugs and drug development, provide strong motivation for new and complementary strategies for research and clinical practice[62]. The network based approaches classifying diseases based on pathways impacted have the potential to substantially enable the elaboration of a network based view of drug discovery and reposition (the application of known drugs to new indications), which are challenging issues in pharmaceutical science[63,64].

Prior to clinical implementation, major challenges must be addressed from a clinician's perspective, including understanding how network approach based genomic science is generated and linked to patient-oriented science, which is the framework for evaluating genomic studies, as an evidence base for providing effective precision medicine to patients in the future[65]. The diseases studied in our work are far more than other researches, not limited to cancers or certain diseases. This fact may lead to our results extensive, however, it may guide people to pay attention to some unexpected points.

A comprehensive description of the associations between pathways and diseases requires identification of not only multiple pathways associated with a specific disease but also pathways associated with multiple diseases. In 9,125 tumors, Sanchezvega et al. point out significant representation of individual and co-occurring actionable alterations in 10 signaling pathways[29]. Meanwhile, we find that most of the 10 signaling pathways are subject to significant influence by diseases, including Hippo signaling pathway, PI3K-Akt signaling pathway, Notch signaling pathway, p53 signaling pathway, cell cycle pathway, Ras signaling pathway, TGF-beta signaling pathway, Wnt signaling pathway. The current understanding of the hippo pathway, for example, has been reviewed with an emphasis on the effects of this pathway on basic biology and human diseases including cancers, immunity and cardiovascular diseases[66]. Another review shows that individuals with RASopathies share many overlapping characteristics, including cardiac malformations, short stature, neurocognitive impairment, craniofacial dysmorphism, cutaneous, musculoskeletal, and ocular abnormalities, hypotonia and a predisposition to developing cancer[67]. These results suggest resemblance between cancer diseases and non-cancer diseases, and implies opportunities for targeted and combination therapies. Integrative analysis methods have been proposed to improve power and reproducibility for identifying genes and prognosis markers associated with multiple cancers, which may lead to discovery of novel therapeutic targets for cancer therapies[68,69]. In addition, it estimates the effect (either positive or negative) of the same gene in different diseases, which helps predict possible side effects of the drug. Supported by our classification of diseases, similar applications can be implemented between cancer diseases and non-cancer diseases to discover biomarker targets and improve drug development for multiple diseases that are classified in the same category.

Also, it is important to have the paper checked for spelling mistakes.

Response:

Thank you very much for your suggestion. We have checked and corrected the spelling mistakes in the manuscript.

Reviewer: 1

Comments to the Author(s)

The manuscript is dedicated to the diseases classification, which is the foundation for achieving precision medicine. Considering that diseases genes affect pathway functions through topology modules, the authors collected data from NCBI and KEGG and carried out diseases classification models from the perspective of gene network modules and pathways. In addition, the authors were trying to perform different classification models of diseases taking account of intensity and scope of the impact of the disease on pathways. The study is solid and the results reported are interesting. After close evaluation the manuscript I have some comments and questions:

1 The authors summarized many databases of different institutions and organizations in the introduction, why not combine all databases but only used data from NCBI and KEGG?

Response:

Thank you very much for your suggestion. In our manuscript, human gene information, gene-gene interactions, disease genes and pathway genes are need. The reason that we get the data from NCBI and KEGG are as follows:

NCBI provides a large suite of online resources for biological information and data, including the GenBank nucleic acid sequence database and the PubMed database. The GenBank nucleic acid sequence database implements daily data exchange through an international collaboration with the DNA Data Bank of Japan (DDBJ) and the European Nucleotide Archive (ENA), so that the core data contained in these databases is the same. In addition to maintaining the GenBank database, NCBI provides many other kinds of biological data as well as retrieval systems and computational resources for the analysis of GenBank and other data. Therefore, we used the human gene information and gene-gene interaction information provided by NCBI to construct the human gene network.

KEGG is a knowledge base for systematic analysis of gene functions, linking genomic information with higher order functional information. The KEGG DISEASE database provides not only disease genes but also its classification of diseases based on congenital and acquired factors as well as diseased tissues and organs. The KEGG PATHWAY database provides human pathways representing our knowledge on the molecular interaction, reaction and relation networks for biological interpretation of higher-level systemic functions. In our manuscript, the genes collection of each disease is regarded as a disease module, and the pathways represent the function modules.

2 *It will make sense if the authors appropriately pointed out good examples of diseases that are grouped together in Figure 4 and Figure 5.*

Response:

Thank you very much for your suggestion. We add a subsection ‘Associations and differences between the two classifications’ in the manuscript. In this part, we give some examples of disease pairs after discussing associations and differences, respectively. And we add a figure (Figure 7) that is an illustration of diseases networks to help to make our results visible.

For example, in the middle of Figure 7(a), it shows that many cancers have common disease genes, and are also classified into one group in our classification. What is more, the giant cell tumor of bone (1699, classified as a musculoskeletal disease in KEGG DISEASE, see supplementary materials), is closely connected with these cancer diseases. Giant cell tumor of the bone is a relatively uncommon tumor of the bone. Malignancy in giant cell tumor is uncommon, however, if malignant degeneration does occur, it is likely to metastasize to the lungs[51]. So it is reasonable to group giant cell tumor of bone with many cancers including non-small cell lung cancer (19) and small cell lung cancer (20).

More examples are available in the subsection ‘Associations and differences between the two classifications’.

3 *Associations and differences between the two classifications should be described in more details.*

Response:

Thank you very much for your suggestion. We add a subsection ‘Associations and differences between the two classifications’ in the manuscript. And we add a figure (Figure 7) that is an illustration of diseases networks to help to make our results visible. Here is a brief summary.

In this paper, we integrate disease genes, topological modules, and functional pathways to assess the influence of diseases to pathways of human body. When it comes to associations between the two classifications, the first is that both of them are based on the impact score, which is the basic metric in this paper (More details can be obtained in ‘the influence on pathways by diseases’ in Methods). The second is the criteria for classifications (More details can be obtained in ‘Classification of diseases’ in Methods and Figure 6). The third is that most disease pairs are grouped together in both of the two classifications (illustrated in gray numbers in Tab.5, as the maximum either in the row or in the column), however, many of them are not in the same group in KEGG DISEASE database (see in Fig.7(a)).

When it comes to differences between the two classifications, the first is that they emphasize the impact of disease on pathways from different perspectives. To classify diseases by the intensity of effects on pathways, we use the normalized vector $IS^N(D_k)$ to measure the difference in intensity between the pathways. The score of the most affected pathway is 1, and the score of the least affected pathway is 0. To classify diseases by the scope of effects on pathways, we use the binary vector $IS^B(D_k)$ to mark pathways with a score exceeding the average. The second is the distance of diseases. In the first classification, the distance of diseases D_i and D_j is defined as

the euclidean distance of $IS^N(D_i)$ and $IS^N(D_j)$. In the second classification, the distance of diseases D_i and D_j is defined as the manhattan distance of $IS^B(D_i)$ and $IS^B(D_j)$. Note that for two binary vectors, The following equation holds $distance_{manhattan} \times dimension_{vector} = (distance_{euclidean})^2$. The third is that there exist some disease pairs are grouped differently in the two classifications (see in Figure 7(b)(c)).

More details and examples are available in the subsection ‘Associations and differences between the two classifications’.

4 How to assess the impact of discovering new genes in the future on current results?

Response:

Thank you very much for your suggestion. We have discussed this question in the section ‘Discussion’, as following: (6th paragraph in Discussion)

The method for deriving the results of this paper is based on the topological structure of gene interaction network especially the largest connected component (LCC) and module division results. After the completion of the Human Genome Project, the number of new genes discovered in the future should be very small. However, it is unavoidable that the topological structure of gene interaction network will be different because of the exploration of new gene-gene interactions. We discussed this question in the ‘Discuss’ section. Here is a brief overview.

We take the published time of the literatures as the time of discovery of interactions, based on which we obtain the LCC of the human gene network at the end of each year (from 2003 to 2017). The number of genes in the LCC increases, however, the number of modules does not change much (more details are available in supplementary materials). This fact may due to properties of scale-free networks that most interactions newly discovered connect a hub gene and an isolated gene. Although the human gene network will inevitably undergo local changes with the discovery of new genes and gene-gene interactions, the overall organization and layout of the network will not be changed significantly, because it is unchanged that hub nodes play a major role in the modules.

5 Line 32 in page 5, “play a role in develop the disease” should be “plays a role in developing the disease”.

Response:

Thank you very much for pointing it out. The mistake has been corrected.

6 Line 37 in page 5, “Selecting this measure mainly because” should be “This measure is selected mainly because”.

Response:

Thank you very much for pointing it out. The mistake has been corrected.

Reviewer: 2

Comments to the Author(s)

This paper proposed several measures to quantify the effect of diseases on gene pathways and functional networks and to quantify the distance between diseases and used these measures to perform hierarchical clustering of diseases. The authors essentially performed unsupervised learning based on gene networks based on diseases information. *The authors may explain in detail how the analysis results especially those based on the disease distance can be useful in precision medicine (as claimed by the authors), in comparison to many developed methods in the literature for predicting disease based on gene information, it is unclear.*

Response:

Thank you very much for your suggestion. We add a figure (Figure 8) to illustrate the relationship between our work and previous works based on gene information, and we have discussed this question in the section 'Discussion', as following: (7th to 9th paragraphs in Discussion)

Disease classification is a progression towards precision medicine with the need for precise patient characterization, currently based on clinical phenotypes but in future augmented by laboratory-based tests[60]. As illustrated in Fig.8, researchers of hospitals and institutions obtain molecular data of patient samples for diseases of concern. Their great studies have been providing valuable guidance to precision medicine from many aspects, such as providing predictive, prognostic, diagnostic, and surrogate markers of diverse disease states, informing on underlying molecular mechanisms of diseases, allowing for classification of patients based on molecular data etc. Our work (the blue part) is an extension on their basis, firstly, because the data we need, for example, disease genes are identified and summarized in previous researches. Secondly, the direct application of genetic information can be considered as first-order, while the human gene network is an integration of genetic information, which is high-order. Thirdly, our classifications of diseases are at a system level, designed to provide novel insights for the clinic practice at the sample level, for example a repositioning of our understanding of diseases and exploration the potential of combination therapy. Also, our results of diseases classifications may complement each other with the classification of complications, as a clear definition of complications is essential in medicine, mainly aiming to improve quality in patients' care. The lack of method for uniform reporting of complications both in terms of definition and grading prompted the authors to propose a classification system of complications based on combining outcome and severity of sequelae[61]. The integration of such work will play a role in guiding combination therapy. Moreover, the enormous complexity of common diseases and the resulting problems, such as the fact that many patients do not respond to treatment and the increasing costs of drugs and drug development, provide strong motivation for new and complementary strategies for research and clinical practice[62]. The network based approaches classifying diseases based on pathways impacted have the potential to substantially enable the elaboration of a network based view of drug discovery and reposition (the application of known drugs to new indications), which are challenging issues in pharmaceutical science[63,64].

Prior to clinical implementation, major challenges must be addressed from a clinician's perspective, including understanding how network approach based genomic science is generated

and linked to patient-oriented science, which is the framework for evaluating genomic studies, as an evidence base for providing effective precision medicine to patients in the future[65]. The diseases studied in our work are far more than other researches, not limited to cancers or certain diseases. This fact may lead to our results extensive, however, it may guide people to pay attention to some unexpected points.

A comprehensive description of the associations between pathways and diseases requires identification of not only multiple pathways associated with a specific disease but also pathways associated with multiple diseases. In 9,125 tumors, Sanchezvega et al. point out significant representation of individual and co-occurring actionable alterations in 10 signaling pathways[29]. Meanwhile, we find that most of the 10 signaling pathways are subject to significant influence by diseases, including Hippo signaling pathway, PI3K-Akt signaling pathway, Notch signaling pathway, p53 signaling pathway, cell cycle pathway, Ras signaling pathway, TGF-beta signaling pathway, Wnt signaling pathway. The current understanding of the hippo pathway, for example, has been reviewed with an emphasis on the effects of this pathway on basic biology and human diseases including cancers, immunity and cardiovascular diseases[66]. Another review shows that individuals with RASopathies share many overlapping characteristics, including cardiac malformations, short stature, neurocognitive impairment, craniofacial dysmorphism, cutaneous, musculoskeletal, and ocular abnormalities, hypotonia and a predisposition to developing cancer[67]. These results suggest resemblance between cancer diseases and non-cancer diseases, and implies opportunities for targeted and combination therapies. Integrative analysis methods have been proposed to improve power and reproducibility for identifying genes and prognosis markers associated with multiple cancers, which may lead to discovery of novel therapeutic targets for cancer therapies[68,69]. In addition, it estimates the effect (either positive or negative) of the same gene in different diseases, which helps predict possible side effects of the drug. Supported by our classification of diseases, similar applications can be implemented between cancer diseases and non-cancer diseases to discover biomarker targets and improve drug development for multiple diseases that are classified in the same category.

The authors should have explained the scientific motivations for defining those quantities and for performing clustering of diseases.

Response:

Thank you very much for your suggestion. We have discussed this question in the section ‘Discussion’, as following: (3rd to 4th paragraphs in Discussion)

It is considered to be right that disease genes affect pathway functions through topology modules[32,33]. Therefore, we derive the influence on pathways by diseases through calculating inner product of the following two vectors. The first vector is used to measure the propagation efficiency of specific disease signals in each of modules. Each component of the vector is the summation of the closeness centrality of disease genes within the corresponding module. Mathematically, the technique of network propagation is simplifying and unifying. It is a powerful data transformation method of broad utility in genetic research, since it greatly improves the power of genetic association, providing a universal amplifier for genetic analysis[56]. The

second vector is a measure of relevance between the module and pathways. Each component of the vector is the jaccard similarity coefficient of the module and corresponding pathway, indicating the extent of overlap of two large gene sets. This measure is selected mainly because it has proven to be doing well in comparing two sets of nodes when considering the difference of the size of the two sets[57]. Moreover, it provides an intuitive way to characterize the set similarity.

Hierarchical clustering has been the dominant approach to constructing classification schemes, and much early work on hierarchical clustering was in the field of biological taxonomy from 1950s and more so from the 1960s onwards[58]. The dendrogram expresses many of the proximity and classificatory relationships in a body of data. To answer the question "how many groups are there?", we defined a parameter to measure the difference of average distances of diseases within groups to average distances of distances between groups, playing the same role as modularity in module partition.

References

- [4] Dozmorov MG. 2018 Disease classification: from phenotypic similarity to integrative genomics and beyond. *Briefings in Bioinformatics*.
- [22] Goh KI, Cusick ME, Valle D, Childs B, Vidal M, Barabási AL. 2007 The human disease network. *Proceedings of the National Academy of Sciences* 104, 8685–8690.
- [29] Sanchez-Vega F, Mina M, Armenia J, Chatila WK, Luna A, La KC, Dimitriadoy S, Liu DL, Kantheti HS, Saghafinia S et al. 2018 Oncogenic Signaling Pathways in The Cancer Genome Atlas. *Cell* 173, 321–337.
- [32] Barabási AL, Gulbahce N, Loscalzo J. 2011 Network medicine: a network-based approach to human disease. *Nature reviews genetics* 12, 56.
- [33] Ravasz E, Somera AL, Mongru DA, Oltvai ZN, Barabási AL. 2002 Hierarchical organization of modularity in metabolic networks. *science* 297, 1551–1555.
- [51] Pai SB, Lalitha R, Prasad K, Rao SG, Harish K. 2005 Giant cell tumor of the temporal bone—a case report. *BMC Ear, Nose and Throat Disorders* 5, 8.
- [56] Cowen L, Ideker T, Raphael BJ, Sharan R. 2017 Network propagation: a universal amplifier of genetic associations. *Nature Reviews Genetics* 18, 551.
- [57] Bass JIF, Diallo A, Nelson J, Soto JM, Myers CL, Walhout AJ. 2013 Using networks to measure similarity between genes: association index selection. *Nature methods* 10, 1169.
- [58] Hennig C, Meila M, Murtagh F, Rocci R. 2015 *Handbook of cluster analysis*. CRC Press.
- [60] Leslie RD, Palmer J, Schloot NC, Lernmark A. 2016 Diabetes at the crossroads: relevance of disease classification to pathophysiology and treatment. *Diabetologia* 59, 13–20.
- [61] Filippiadis D, Binkert C, Pellerin O, Hoffmann R, Krajina A, Pereira P. 2017 Cirse quality assurance document and standards for classification of complications: the cirse classification system. *Cardiovascular and interventional radiology* 40, 1141–1146.
- [62] Benson M. 2016 Clinical implications of omics and systems medicine: focus on predictive and individualized treatment. *Journal of internal medicine* 279, 229–240.
- [63] Schadt EE, Friend SH, Shaywitz DA. 2009 A network view of disease and compound screening. *Nature reviews Drug discovery* 8, 286.
- [64] Iwata H, Sawada R, Mizutani S, Yamanishi Y. 2015 Systematic drug repositioning for a wide range of diseases with integrative analyses of phenotypic and molecular data. *Journal of chemical information and modeling* 55, 446–459.
- [65] Gaziano JM, Concato J, Brophy M, Fiore L, Pyarajan S, Breeling J, Whitbourne S, Deen J, Shannon C, Humphries D et al. 2016 Million Veteran Program: a mega-biobank to study genetic influences on health and disease. *Journal of clinical epidemiology* 70, 214–223.
- [66] Ma S, Meng Z, Chen R, Guan KL. 2018 The Hippo pathway: Biology and pathophysiology. *Annual review of biochemistry*.
- [67] Dard L, Bellance N, Lacombe D, Rossignol R. 2018 RAS signalling in energy metabolism and rare human diseases. *Biochimica Et Biophysica Acta (BBA)-Bioenergetics* 1859, 845–867.
- [68] Ma S, Huang J, Moran MS. 2009 Identification of genes associated with multiple cancers via integrative analysis. *BMC genomics* 10, 535.
- [69] Ma S, Huang J, Wei F, Xie Y, Fang K. 2011 Integrative analysis of multiple cancer prognosis studies with gene expression measurements. *Statistics in medicine* 30, 3361–3371.